# Feature-Learning Networks Are Consistent Across Widths At Realistic Scales

**Nikhil Vyas**[1*]    **Alexander Atanasov**[2,3,4*]    **Blake Bordelon**[1,3,4*]
**Depen Morwani**[1,3]    **Sabarish Sainathan**[1,3,4]    **Cengiz Pehlevan**[1,3,4]
[1]SEAS    [2]Department of Physics    [3]Kempner Institute    [4]Center for Brain Science
Harvard University
{nikhil,atanasov,blake_bordelon,dmorwani,
sabarish_sainathan,cpehlevan}@g.harvard.edu

## Abstract

We study the effect of width on the dynamics of feature-learning neural networks across a variety of architectures and datasets. Early in training, wide neural networks trained on online data have not only identical loss curves but also agree in their point-wise test predictions throughout training. For simple tasks such as CIFAR-5m this holds throughout training for networks of realistic widths. We also show that structural properties of the models, including internal representations, pre-activation distributions, edge of stability phenomena, and large learning rate effects are consistent across large widths. This motivates the hypothesis that phenomena seen in realistic models can be captured by infinite-width, feature-learning limits. For harder tasks (such as ImageNet and language modeling), and later training times, finite-width deviations grow systematically. Two distinct effects cause these deviations across widths. First, the network output has an initialization-dependent variance scaling inversely with width, which can be removed by ensembling networks. We observe, however, that ensembles of narrower networks perform worse than a single wide network. We call this the *bias* of narrower width. We conclude with a spectral perspective on the origin of this finite-width bias.

## 1 Introduction

Studies of large-scale language and vision models have shown that models with a larger number of parameters achieve better performance [1, 2]. Motivated by the success of large-scale models, several theories of deep learning have been developed, including large-width limits. Infinite width limits which arise in standard parameterization (SP) or neural tangent parameterization (NTP) considered in [3, 4] gives rise to a a model with no feature learning (a kernel method). In this limit, the neural network loses the ability to adapt its internal features. Feature learning is crucial to explain deep learning's superior performance to kernels, the emergence of interpretable neurons such as edge-detecting CNN filters, transfer learning capabilities, and large learning rate effects such as edge of stability [5–8]. All of these effects are exhibited in modern large-scale networks.

Recently, several works have identified an alternative parameterization of neural networks that preserves feature-learning even at infinite width [9–14]. In this work, we focus on the maximal update parameterization ($\mu$P), or equivalently the mean field parameterization [12, 15, 14]. The existence of infinite-width feature-learning limit, suggests that this parameterization is potentially more promising to explain deep learning phenomena than the previous limits. This motivates us to ask:

**Question:** *Can realistic-width neural networks be accurately described by their infinite-width feature-learning limits?*

---

*These authors contributed equally to this work.

37th Conference on Neural Information Processing Systems (NeurIPS 2023).

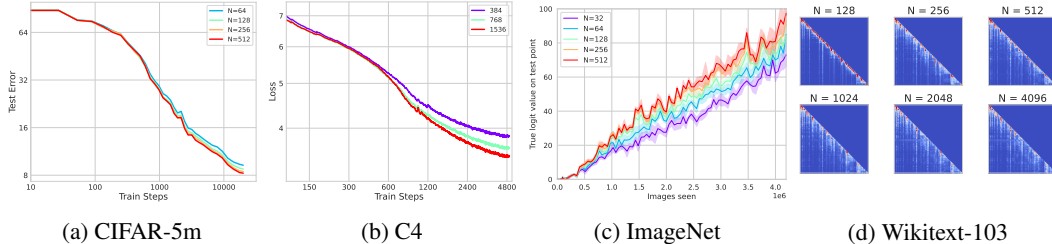

| (a) CIFAR-5m | (b) C4 | (c) ImageNet | (d) Wikitext-103 |

Figure 1: Consistency of large width behavior across tasks, architectures, observables. a) Loss curves for Resnets on Cifar-5M in $\mu$P are nearly to identical at large widths (see also Figure 2). b) For GPT-2 on the C4 dataset [16] the loss curves agree at early times and deviate at late times, but wider networks agree for longer (see also Figure 2 and appendices for Wikitext-103) c) The values that ResNets put on the correct logit for ImageNet appear to converge as the width grows (see also Figure 3). d) The attention matrices for transformers on Wikitext-103 become nearly identical as width increases (for quantitative metrics see Figure 5.)

We attempt to answer this question by training networks of varying widths on vision and language tasks for realistic datasets and architectures. Concretely, we focus on on the online setting, where data is not repeated during SGD, and track the following quantities across widths:

- The losses throughout training.
- The predictions of the networks on individual points throughout training.
- The learned representations, summarized by the feature kernels preactivation distributions and, for transformers, their attention matrices.
- Dynamical phenomena such as the edge of stability governing the top Hessian eigenvalues, as well as large learning rate and small batch size effects on the loss.

On each of these metrics, we show that sufficiently wide neural networks converge to consistent behavior across widths. In Figure 1, we show loss curves, logit predictions, and attention matrices approach consistent behavior as width is increased across several architectures and datasets. We further observe that the widths that achieve this consistent behavior are within the range of those used in practice. We use large-width consistency as a proxy for achieving the limiting infinite-width behavior. We stress that this observed consistency is a property of networks in mean field/$\mu$P parameterization but is not present in other parameterizations which also give an infinite width limit like NTK parameterization (See Appendix D for a comparison).

We say that a network property is consistent if, beyond some width, its values all lie within some small interval with high probability. We measure consistency by showing that a quantity's deviations between successive widths decrease as the widths are increased, and that its value for narrower networks systematically approaches its value for the largest trained network.

Our results show the following:

- For simple vision tasks such as CIFAR-5m [17], ResNets with practical widths achieve near consistent loss curves across widths (Section 2).
- Beyond the loss curves, the individual predictions of the networks agree pointwise. That is, the logits agree on test points throughout the training process. We further show that internal representations as measured by distributions of neuron preactivations and feature kernels in various layers are consistent across widths (Section 2).
- For harder tasks such as ImageNet and language modeling, loss curves are consistent across widths early in training. As training progresses, loss curves for narrow networks deviate smoothly from the loss curves of wider networks. The effective width required to reach infinite-width behavior thus increases with training time. Conversely, as network size grows we approximate the infinite width network for a larger number of training steps (Section 2).
- Finite-width neural networks have variance in the learned function due to initialization seed. This variance depends inversely on the width. We study ensembles of networks over different initializations to remove this noise. Further, by training ensembles of networks, we can perform

a bias-variance decomposition over initializations (c.f. Appendix F for details and definitions) to analyze the effects of finite width. We find that finite-width bias plays an important role. Equivalently, ensembling narrow networks does not yield infinite-width behavior (Section 3).

- In the setting of offline learning, at late times one can over-fit the training set. We observe that this leads to larger gaps in network behavior across widths, and can break the trend that wider networks perform better (Section 3).

- We develop a spectral perspective on the origin of the finite-width bias by analyzing it in a simple setting of a lazy network learning a simple task. We then apply this perspective to a CNN trained on CIFAR-5m (Section 4).

The consistency across large widths strongly suggests that the dynamics and predictions of realistic-scale networks can be effectively captured by their infinite-width feature learning limits. For realistic tasks, as the width is increased, a larger interval of training can be characterized by this infinite-width limit. Most importantly, even though quantitative agreement across widths slowly breaks with more and more training, we observe that wider networks perform better (as in [15]) and preserve qualitative aspects of the learned features (such as hidden layer kernels and attention matrices) and dynamical phenomena such as edge of stability. **This suggests that infinite width feature-learning networks are good models to study deep learning.**

Our results have implications for interpretability, as the agreement of internal representations suggest that many other phenomena, such as transfer learning with linear probes or fine-tuning, in-context learning [18, 19], the emergence of outliers [20], and the emergence of induction heads [21] may be understood from the perspective of infinite-width feature learning networks.

We plan to have our code made freely available on github to ensure the reproducibility of these results.

## 1.1 Related Works

Empirically, the scaling of relevant quantities with width in the standard or neural-tangent parameterizations was thoroughly studied in [22]. In the latter parameterization, sufficiently wide networks give a kernel method with the infinite-width NTK. Several papers have shown that in practice the NTK limit insufficiently characterizes realistic deep neural networks [5, 23, 6]. Attempts to capture feature learning and predictor variance from perturbative series around infinite-width dynamics show that finite-width variance and kernel adaptation scale as $1/N$ [24–26] for width $N$. A $1/N$ scaling of generalization error with width was empirically verified on many tasks [27, 28]. The effect of width on generalization in the feature-learning regime was empirically studied in [29] in the relatively limited setting of multi-layer perceptrons (MLPs) on polynomial tasks. There, the variance of the finite-width NTK at the end of training adversely affected generalization. Bias-variance decompositions over dataset, label noise, and initialization parameters were studied in [30] for linear models.

The authors of [31] identified that altering the output scale $\alpha$ of any network could increase or decrease feature learning in a neural network. Large values of $\alpha$ correspond to the "lazy limit" where the network's features don't evolve. A follow up study noticed that rescaling the output by $\alpha = \alpha_0/\sqrt{N}$ for width $N$ networks gave consistent behavior of feature learning and losses in small scale experiments [10]. Several works have studied this regime of training in the two-layer limit, known as "mean field" parameterization, where features are still learned even at infinite width [32, 9, 33, 34]. Extensions of this model to deeper networks were studied in [35–38, 12, 14]. A theory of finite-width corrections to networks in this parameterization was studied in [39]. A very general set of parameterization principles, termed $\mu$P, was introduced to give a well defined feature learning limit for a wide range of architectures including RNNs, CNNs, MLPs and transformers [12]. [15] demonstrated that this parameterization nearly fixes optimal hyperparameters across network widths, allowing for hyperparameter transfer from small to large widths. This work also empirically noted that wider networks always outperformed narrower networks in this parameterization.

Our paper focuses on networks in $\mu$P and attempts to study the consistency of many relevant network properties across widths. We perform a fine-grained analyses of more realistic models throughout the dynamics of training. To the best of our knowledge, this is the first such paper to study the consistency of network outputs, internal representations, and dynamics across widths.

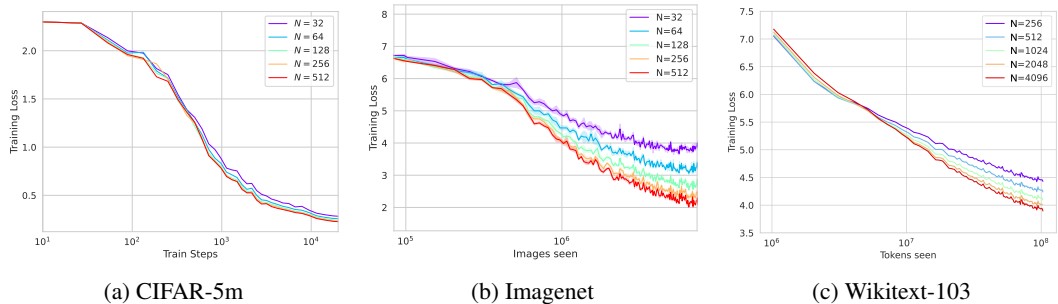

(a) CIFAR-5m        (b) Imagenet        (c) Wikitext-103

Figure 2: In the online learning setting, train loss improves as width grows. For sufficiently wide networks, the training lost is consistent across widths. For Cifar-5m this consistency is observed over all of training. For harder tasks like Imagenet and Wikitext-103, networks of different widths agree up until a width-dependent time-step where narrower networks begin performing worse.

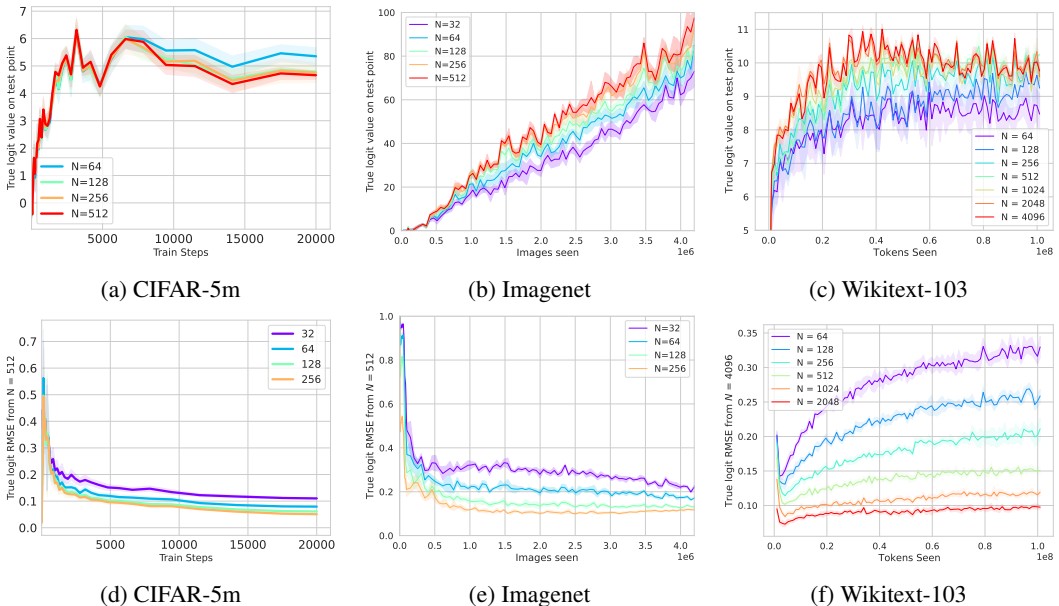

(a) CIFAR-5m        (b) Imagenet        (c) Wikitext-103

(d) CIFAR-5m        (e) Imagenet        (f) Wikitext-103

Figure 3: The output logits on a fixed test point diplays stable behavior at large enough widths. a) Value of network on correct class logit over time as width is varied for CIFAR-5m. Colored errorbars represent one standard deviation. b) Same plot for Imagenet for a fixed image in the test set c) Same plot for Wikitext-103 for a fixed masked token. Across the board the widest networks behave similarly. Next, we use the widest network as a proxy for the infinite-width limit, and compare the logit predictions of narrower networks against that. d) For CIFAR-5m, the relative root-mean-squared error over the test set of the distance to the value that the widest network puts on the correct logit. e) The same for Imagenet. f) The same for Wikitext-103. We see a striking regularity of networks converging to the widest one as the width grows. In Appendix B, we also compare networks of successive widths and show the the difference shrinks.

## 2 Consistency of large-width behavior in online learning

We focus on studying the effect of width in the setting of neural networks learning a task in the online setting. Online learning is representative of many modern settings of deep learning, and as will be shown in Section 3, obviates consideration of memorization and over-fitting in offline learning that can lead to large differences in networks across widths.

In what follows, the variable $N$ will denote the width of a given network. For vision tasks, this will correspond to the number of channels in each layer. For transformers, in the notation of [40], $N = d_{model} = hd_k = hd_v$ and $d_{ffn} = 4N$. Here, $h$ is the number of heads, which we will keep

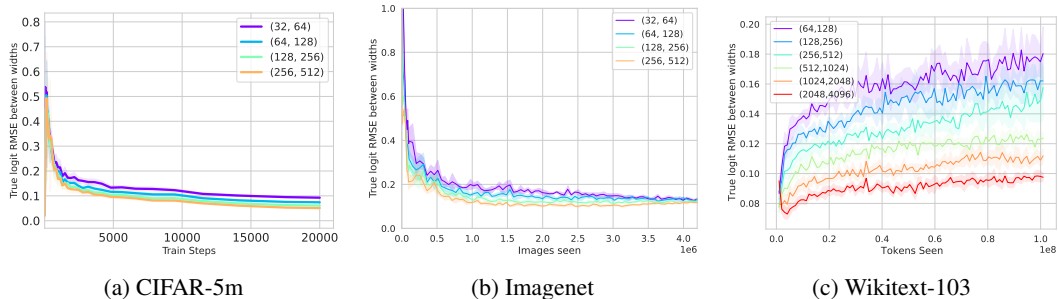

(a) CIFAR-5m     (b) Imagenet     (c) Wikitext-103

Figure 4: Analog of the last row of Figure 3 but comparing networks of successive widths rather than comparing all networks to the widest. Again, we see that as the network width grows, the difference between successive networks shrinks.

fixed. $d_{model}$ is the embedding dimension of the tokens as well as the dimension of the residual stream. $d_k$ is the dimension over which the dot products in the attention are calculated and $d_v$ is the dimension of the values in the attention layers. $d_{ffn}$ is the hidden width of the feedforward networks (FFN).

**Convergence of loss curves**   We begin by showing (Fig. 2) that the loss curves for sufficiently wide networks on a given task achieve consistent behavior across widths. Throughout the paper we measure train loss in terms of crossentropy. For all tasks, at early times large widths agree, but for more complicated tasks such as ImageNet or Wikitext-103, learning curves of narrower network deviate from those of wider ones.

The width beyond which networks emulate infinite-width behavior depends on the complexity of the task. For more difficult tasks, larger widths are required for the loss curves to converge. For simple tasks such as CIFAR-5m we find that widths as narrow as 128 are essentially consistent with infinite width-behavior for an entire pass through the 5 million image dataset. For ImageNet, widths near 512 are close to consistent for four passes through the dataset with heavy data augmentation. These widths are well within the range of those practically for images [41, 42]. For transformers going through a single full pass of Wikitext-103, widths on the order of 4000 are required. Early transformer models certainly had hidden widths of order 4k [43], and more recent models such as GPT-3 have widths going up to 12288 [18], so this is also within the regime of realistic width.

**Pointwise convergence of predictions**   Beyond the convergence of the training loss curves, we observe that the logits of a network on a fixed test point become consistent as width grows. This test point can be an image in the test set or a masked token in the validation set. In plots a), b), and c) of Figure 3, we show that for a specific held-out test point, the value of the network on the correct logit becomes consistent as the width grows. In d), e), and f) we plot the root mean squared distance to the widest networks logits over the test set. We further study the difference between successive widths in Figure 4.

**Convergence of representations**   In addition to loss and prediction dynamics, we also examine whether learned representations in these models are consistent across widths. Mean field theories of neural network dynamics predict that sufficiently wide networks should have identical kernels (and attention matrices for transformers) and that all neurons in a layer behave as independent draws from an initialization-independent single-site distribution [9, 12, 14, 46, 47]. To test whether realistic finite-width feature learning networks are accurately captured by this limit, in Figure 5, we analyze the feature kernels and preactivation distributions before and after training as well as the attention matrices in transformer models trained on Wikitext-103. We see qualitative consistency in the plots of kernels and attention matrices in b) and c) which can be made quantitatively precise by plotting the distance to the widest networks and showing systematic convergence in c) and f).

**Convergence of dynamical phenomena**   In Figure 6a, we show that the sharpness, defined as the top eigenvalue of the loss Hessian, grows steadily to a final value that it then fluctuates around. This is a small-batch analog of the the edge-of-stability phenomenon identified in [7]. We also show in Figure 6b that on CIFAR-5m task, at early times, the individual variations due to batch noise and large learning rate effects can be consistently captured across widths for $\mu$P networks. In Appendix

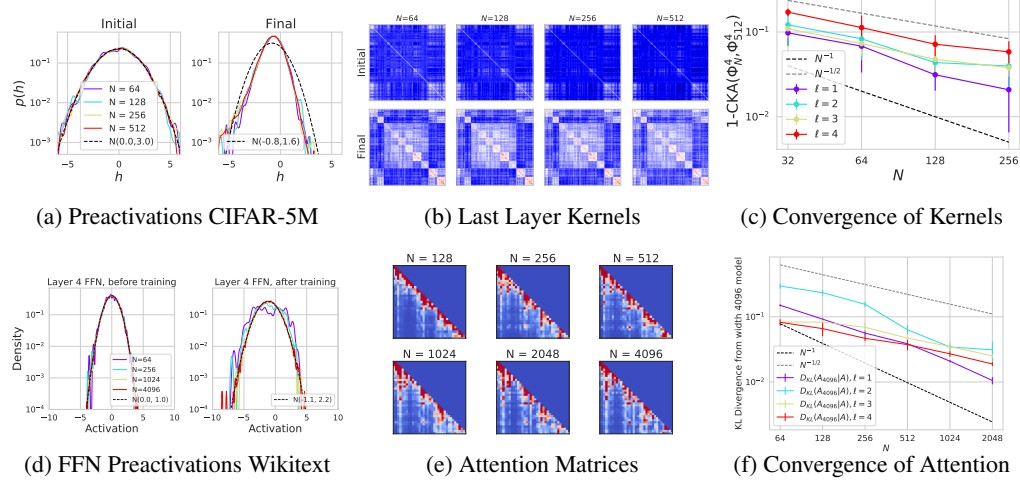

(a) Preactivations CIFAR-5M     (b) Last Layer Kernels     (c) Convergence of Kernels

(d) FFN Preactivations Wikitext     (e) Attention Matrices     (f) Convergence of Attention

Figure 5: Learned features are consistent across a large range of widths in realistic tasks. (a) The distribution (over neurons) of preactivation values $h$ in the final block of $E = 8$ ResNet18 networks trained on CIFAR-5M. At initialization, the densities are all well approximated by the Gaussian with matching mean and variance (dashed black). After feature learning, the density has shifted and become non-Gaussian (poor match with dashed black), yet is still strikingly consistent across widths. (b) Average (over random init) feature kernels are also consistent across widths. (c) The centered kernel alignment CKA [44, 45] of the width $N$ and width $512$ kernels increases towards $1.0$ as $N$ increases. The $1/\sqrt{N}$ and $1/N$ trends are plotted for reference. (d) The preactivation histogram for a transformer on Wikitext-103. At initialization the Gaussian of best fit is the standard normal. After training the histograms are still quite Gaussian, with different moments. (e) A variant of Figure 1 (d) at a smaller sequence length. Attention matrices are consistent at large widths. f) Both FFN kernels and attention matrices converge as width grows. The $1/N$ and $1/\sqrt{N}$ trends are plotted for reference.

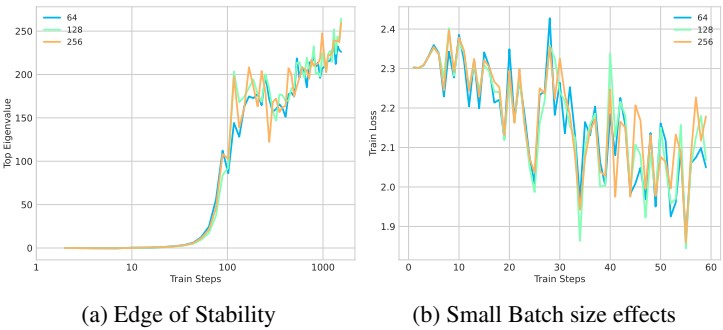

(a) Edge of Stability     (b) Small Batch size effects

Figure 6: Convergence of dynamical phenomena across width for CIFAR-5m

E, we further demonstrate sharp agreement of large learning rate and small batch size phenomena for MLPs learning a simple task. There, we show that while $\mu$P leads to strikingly consistent loss curves, SP does not.

## 3 Deviations from large-width behavior

The consistency observed in Section 2 may break later during training in either the online or offline settings. In the online setting, deviations owing to narrow width compound over time and lead to two sources of error relative to the infinite width limit which we describe in 3.1. In the offline setting, where data is recycled several times, networks over-fit the training data, which can lead to larger gaps between widths and can break the trend that wider networks perform better.

Finite-width effects introduce an initialization dependence to the network, leading to additional variance in the learned function and hindering generalization [27–29]. This initialization-dependent variance can be mitigated by averaging the output logits of a sufficiently large ensemble of networks [48]. Using the bias-variance decomposition terminology, we refer to the discrepancy in performance between an ensembled network and the expected performance of a single network the *variance*, and the gap between an ensembled network and the behavior of infinite-width network as the bias of narrower width. We elaborate thoroughly on what we mean by this decomposition in Appendix F. By definition, the expected difference in loss between a single finite-width network and an infinite-width network is the sum of the bias and the variance. Below, we investigate the behavior of bias and variance in networks across various vision and language tasks.

## 3.1 Online training

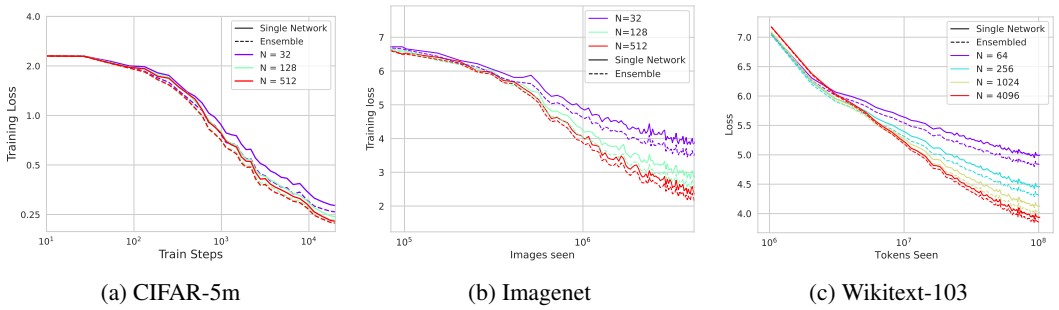

|          (a) CIFAR-5m          |          (b) Imagenet          |          (c) Wikitext-103          |

Figure 7: Loss curves and their ensembles in the online setting. Ensembling reduces the training loss, but a large ensemble of narrow networks do not achieve the performance of a single wider network. Errorbars representing standard deviation over random init are not visible.

Figure 7 shows that at large widths, both single networks and ensembles of networks achieve comparable error. In this regime, all the networks are consistent and increasing the width has a very marginal effect, as does ensembling. At narrower widths, variance is nontrivial (i.e. ensembling helps) but bias is much larger than variance. Single wide networks outperform ensembles of narrower networks. By comparing a) with b) and c) of Figure 7, we see that harder tasks induce larger bias gaps. Prior theoretical work [28, 29] has focused mostly on studying the variance term. In Section 4 we study the bias from a theoretical perspective.

## 3.2 Offline Training

In offline learning, which refers to multi-epoch training, we encounter several unexpected phenomena that challenge the width consistency observed in the previous section, even at large widths. To compare offline learning with online learning, we utilize CIFAR-200k, a 200k sized random subset of CIFAR-5m. Previous studies have demonstrated that label noise contributes to an increase in overfitting [49]. In order to investigate how width consistency changes with overfitting and double descent, we conduct experiments on a noisy label version of CIFAR-50k (50k sample from CIFAR-5m), where 50% of the labels are noisy. Additional ImageNet experiments are presented in Appendix J. As offline training achieves near-zero error, we need to compare very small quantities. To accomplish this, we will plot and compare all quantities on a logarithmic scale. The following phenomena are observed:

- Single network performance on the training set does not converge with width, even at high widths (Figure 8 (a)). In other words, the combined bias and variance does not reach zero, even with substantial widths. This is in contrast to the online runs.

- Ensembling (Figure 8 (b)) reveals that both bias and variance terms individually fail to reach zero, even at high widths.

- Regarding test performance, both bias and variance tend to zero as width increases, demonstrating an instance of benign overfitting (Figure 8 (d) and (e)).

- When working with the noisy label version of CIFAR-50k, we observe clear overfitting and stepwise double descent [49] as training progresses (Figure 8 (f)). Notably, we observe significant

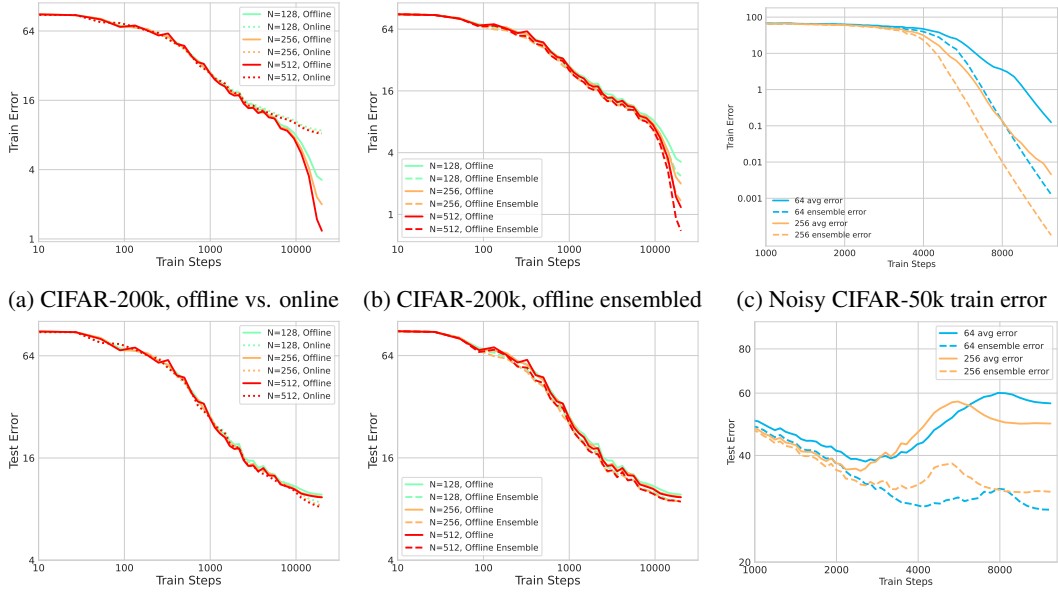

(a) CIFAR-200k, offline vs. online    (b) CIFAR-200k, offline ensembled    (c) Noisy CIFAR-50k train error

(d) CIFAR-200k, offline vs. online.    (e) CIFAR-200k, offline ensembled    (f) Noisy CIFAR-50k test error

Figure 8: Top Row: Effects of offline training on train metrics. In (a) and (b) we do multi-epoch training on CIFAR-200k. We see that both bias and variance for train error are magnified by offline training and do not tend to 0 for the largest widths we could try. (c) we do multi-epoch training on noisy CIFAR-50k and again observe large bias and variance terms at large widths. Bottom Row: Effects of offline training on test metrics. In (d) and (e) we do multi-epoch training on CIFAR-200k. We see that both bias and variance for test error are near 0 at large widths. (f) We train on noisy CIFAR-50k and observe that "wider is better" is violated for ensembled networks.

    deviations in width for single network performance, indicating that the benign overfitting observed in Figure 8 (d) and (e) is dataset-dependent. Furthermore, variance is found to be much larger than in the non-noisy experiments.

- Surprisingly, we discover (Figure 8 (f)) that some ensembled narrower width networks outperform ensembled wider networks. This presents a counterexample to the "wider is better" phenomenon [15] for ensembled networks. We hypothesize that such counterexamples can only exist in the context of offline training.

## 4   Spectral perspective on finite-width bias

In this last section, we develop a toy model in which the effect of finite-width bias can be clearly seen. We analyze it first in the simple setting of an MLP fitting a polynomial in the lazy limit. Here, all the dynamics are well-captured by the finite-width empirical neural tangent kernel (eNTK). By studying the spectral properties of this kernel across widths, we see that finite widths lead to eNTK's with worse bias components in their losses.

Concretely, we see that although the eigenvalue spectrum of the ensembled eNTK is not substantially affected by finite width, the decomposition of the task into eNTK eigenvectors changes, with narrow-width eNTK's putting more of the task into smaller eigenmodes that take longer to be learned. In practice, applying this analysis to the after-kernel of the trained ResNets on CIFAR-5m reveals similar behavior. Prior literature has demonstrated that many of the properties of the final learned function are captured by the after-kernel [50–52].

We consider a model of online learning where a large batch of data from the population distribution $p(\boldsymbol{x})$ is sampled at each step. This leads to approximate gradient flow dynamics $\frac{d}{dt}\boldsymbol{\theta} = -\frac{1}{2}\nabla_\theta \mathbb{E}_{\boldsymbol{x}}(f(\boldsymbol{x}, \boldsymbol{\theta}) - y(\boldsymbol{x}))^2$ (Appendix H). To analyze this equation, we choose a fixed orthonormal basis $\{\psi_k(\boldsymbol{x})\}$ for the space $L^2(\mathbb{R}^D, p(\boldsymbol{x})d\boldsymbol{x})$ of square-integrable functions on input space. The function $f(\boldsymbol{x})$, residual error $\Delta(\boldsymbol{x}) = y(\boldsymbol{x}) - f(\boldsymbol{x})$, and the empirical NTK

$K(\boldsymbol{x}, \boldsymbol{x}', t)$ can be expressed in this basis as $f(\boldsymbol{x}, t) = \sum_k f_k \psi_k(\boldsymbol{x})$, $\Delta(\boldsymbol{x}, t) = \sum_k \Delta_k \psi_k(\boldsymbol{x})$, and $K(\boldsymbol{x}, \boldsymbol{x}', t) = \sum_{k\ell} K_{k\ell}(t)\psi_k(\boldsymbol{x})\psi_\ell(\boldsymbol{x}')$, respectively. Their training evolution is given by:

$$\frac{d}{dt} f(\boldsymbol{x}, t) = \mathbb{E}_{\boldsymbol{x}' \sim p(\boldsymbol{x})} K(\boldsymbol{x}, \boldsymbol{x}', t)\Delta(\boldsymbol{x}', t) = -\sum_{k\ell} K_{k\ell}(t)\Delta_\ell(t)\psi_k(\boldsymbol{x}). \qquad (1)$$

The statistics of the dynamical NTK matrix $K_{kl}(t)$ summarizes the statistics of the error dynamics $\Delta(\boldsymbol{x}, t)$ at any level of feature learning. At infinite width, $K_{k\ell}(t)$ is deterministic, while at finite width, it receives a $\Theta(N^{-1})$ mean displacement and a $\Theta\left(N^{-1/2}\right)$ fluctuation around its mean [24, 25, 39]. We consider approximating the dynamics of the ensembled predictor by $\frac{d}{dt} \langle f_k(t) \rangle_{\theta_0} \approx \sum_\ell \langle K_{k\ell}(t) \rangle \langle \Delta_\ell(t) \rangle$. Here, $\langle \cdot \rangle$ denotes averages over initializations. This expression neglects the contribution from $\mathrm{Cov}(K_{k\ell}, \Delta_\ell)$. We show that this approximation is accurate in depth-3 MLPs trained on Gegenbauer polynomial regression tasks in Figure 9 a). For more details see Appendix A.

In the lazy limit, the kernel is static and we choose $\psi_k$ to diagonalize $\langle K_{k\ell} \rangle = \delta_{k\ell}\lambda_k$. This yields the loss dynamics $\mathcal{L}(t) = \sum_k \langle y(\boldsymbol{x})\psi_k(\boldsymbol{x}) \rangle^2 e^{-2\lambda_k t}$. We can therefore quantify alignment of eigen-functions to task with the cumulative power distribution $C(k) = \sum_{\ell < k} \langle y(\boldsymbol{x})\psi_\ell(\boldsymbol{x}) \rangle_{\boldsymbol{x}}^2 / \langle y(\boldsymbol{x})^2 \rangle_{\boldsymbol{x}}$, which is the proportion of the task that is explained by the first $k$ eigenvectors [53]. If $C(k)$ rises rapidly with $k$ then the loss falls faster [53]. In this limit, there are two ingredients that could make the bias dynamics across widths distinct. First, the eigenvalues $\lambda_k$ which set the timescales could be width-dependent. Second, the eigenfunctions $\psi_k(\boldsymbol{x})$ that diagonalize $\langle K \rangle$ can change with width. In Figures 9 b) and c) we show that the dominant effect is the latter. Finite-width corrections do not substantially affect the spectrum, but they do increase the proportion of the target function that lies in the eigenspaces corresponding to modes that are slower to learn.

To test whether these findings continue to hold in more realistic experiments, we computed the final NTKs (after kernels) of the ResNet-18 models trained on CIFAR-5M (specifically the models from Figures 3, 5). We ensemble average to get kernel $\langle K_{c,c'}(\boldsymbol{x}, \boldsymbol{x}') \rangle$ for output channels $c, c'$ and input images $\boldsymbol{x}, \boldsymbol{x}'$. We then compute the kernel gradient flow corresponding to MSE training on the true target function for CIFAR-5M $\frac{d}{dt}\Delta_c(\boldsymbol{x}) = -\sum_{c'} \mathbb{E}_{\boldsymbol{x}'} \langle K_{c,c'}(\boldsymbol{x}, \boldsymbol{x}') \rangle \Delta_{c'}(\boldsymbol{x}')$ from initial condition given by the one-hot target labels $\Delta_c(\boldsymbol{x})|_{t=0} = y_c(\boldsymbol{x})$. The convergence rate of this dynamical system is again set by the eigenvalues and eigenfunction-task alignment. In Figure 9 (d), we find that the after kernels for wider networks give slightly more rapid convergence. Figures 9 (e) and (f) show that, similar to the MLP experiment, the spectra are very consistent across widths, but the eigenfunction task alignments, measured with $C(k)$ are not. Overall, these experiments suggest that an important aspect of the bias of finite width models compared to their infinite width analogs is the deformation of their eigenfunctions.

## 5    Conclusion

We have demonstrated a striking consistency across widths for many quantities of interest to deep learning practitioners. Our fine-grained studies go beyond simply comparing test losses and have demonstrated that learned network functions, internal representations, and dynamical large learning rate phenomena agree for sufficiently large widths on a variety of tasks across vision and language. At later training times, or after many repetitions of the dataset, we observe systematic deviations brought on by finite width, and have characterized them in terms of the bias and variance of the network over initializations. This study motivates the applicability of infinite-width feature-learning models (and the accumulating finite width deviations from this limit) in reasoning about large scale models trained on real-world data.

In light of the accumulation of finite-width deviations at later training times, we caution that our study only exhibits the consistency of the infinite-width limit with the training time *held fixed*. It does not make a claim about other possibly limits that vary the training time jointly with the width to infinity, perhaps along a compute frontier. We leave further inquiry into such limits for future work.

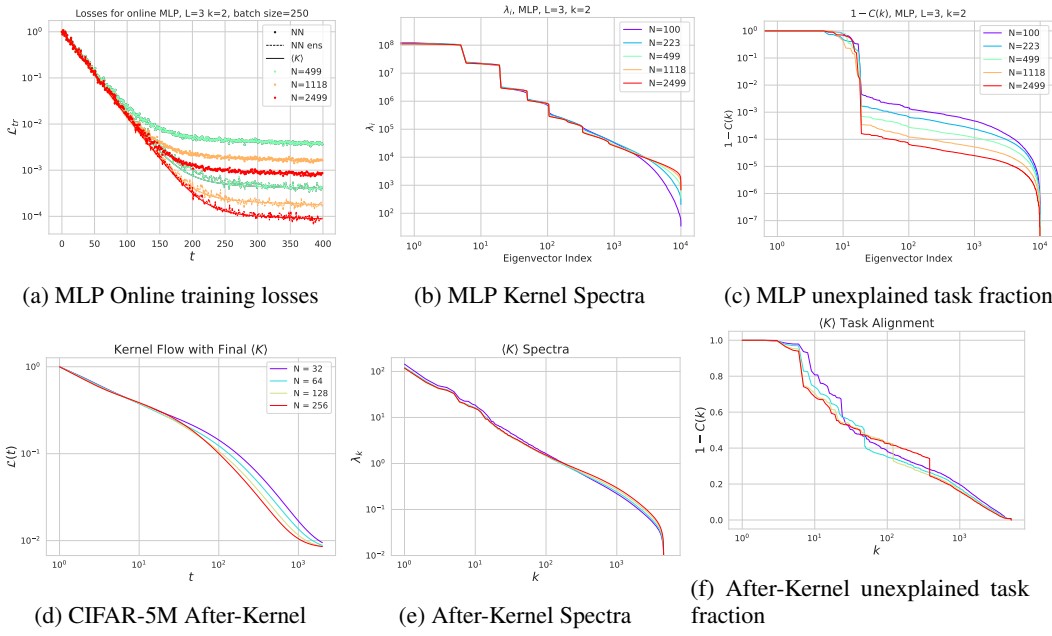

(a) MLP Online training losses     (b) MLP Kernel Spectra     (c) MLP unexplained task fraction

(d) CIFAR-5M After-Kernel     (e) After-Kernel Spectra     (f) After-Kernel unexplained task fraction

Figure 9: Spectral properties of the NTK can account for bias gaps across widths. (a) Depth 3 MLPs in the lazy limit ($\gamma_0^{-1} = 200$) learning a quadratic polynomial from a uniform distribution on the sphere in $D = 5$ dimensions online. Wider networks perform better (dots). Even after ensembling (dashed), wider is better, and the ensembled curves match those of the averaged eNTK (solid). (b) The spectra of the averaged eNTK across widths do not show substantial variability. (c) However, at narrower width, the eigen-decomposition of the task has greater weight along higher spectral modes, consequently leading to a slow-down in training. These results hold across dimensions, batch sizes, task complexity, and architectures. Strong feature learning can reduce this effect. See Appendix H (d) We computed the ensemble averaged *after kernels* from the CIFAR-5M ResNet-18 models and computed the theoretical kernel flow on the task. Wider models have a slightly better mean kernel for this task. (e) The eigenvalues of the final NTKs are very consistent across widths. (f) The eigenfunction-target alignment of the final kernels noticeably differ across widths, evidenced by the cumulative power distribution $C(k)$ which accounts for the gap in theoretical loss curves under kernel flow.

## Acknowledgments and Disclosure of Funding

We thank Boaz Barak, Jeremy Cohen, Alex Damian, Nikhil Ghosh, Gal Kaplun, Eric Michaud, Jamie Simon and Jacob Zavatone-Veth for helpful discussions throughout this project. We also thank Jacob Zavatone-Veth for comments on the draft.

AA is supported by the Professor Yaser S. Abu-Mostafa Fellowship from the Fannie and John Hertz Foundation. NV and DM are supported by a Simons Investigator Fellowship, DARPA grant W911NF2010021,and DOE grant DE-SC0022199. DM is supported by funding from the Office of Naval Research under award N00014-22-1-2377 and the National Science Foundation Grant under award CCF-2212841. SS is supported by a Susan Wojcicki and Dennis Troper Graduate Fellowship. BB is supported by a Google PhD fellowship. NV, BB, DM and CP are supported by funding from NSF grant DMS-2134157. CP is also supported by NSF CAREER Award IIS-2239780, and a Sloan Research Fellowship. This work has been made possible in part by a gift from the Chan Zuckerberg Initiative Foundation to establish the Kempner Institute for the Study of Natural and Artificial Intelligence. Compute was provided by the Harvard FASRC cluster and the Kempner Institute.

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

# A Experimental Details

## A.1 MLPs

In Figure 9 we used a 3-layer MLP learning a Gegenbauer polynomial $Q_2(\boldsymbol{\beta} \cdot \boldsymbol{x})$ in $D = 5$ dimensions. Here $\boldsymbol{\beta}$ was a randomly chosen unit vector in $\mathbb{R}^D$. We implemented $\mu$P parameterization by hand. The output layer of the network was rescaled by $\alpha_0 / \sqrt{N}$, consistent with $\mu$P. We chose $\alpha_0 = 1000$ to put us in the lazy regime. We set the the learning rate to be $5N/(1 + \alpha_0^2)$. General arguments based on kernel scale indicate that the learning rate should be scaled as $\alpha_0^{-2}$ at large $\alpha$.

In Figure 14 we used a 3-layer MLP learning a Gegenbauer polynomial $Q_2(\boldsymbol{\beta} \cdot \boldsymbol{x})$ in $D = 25$ dimensions. We set the learning rate to be nearly as high as possible before a loss explosion.

## A.2 Vision

### A.2.1 CIFAR-5m

All plots except Figure 6a and 6b: We trained with standard CIFAR data augmentation of random crop (RandomCrop(32, padding=4) in pytorch) and horizontal flip (RandomHorizontalFlip() in pytorch). As base network (for $\mu$P) we used ResNet18 where BatchNorm was replaced with LayerNorm (to maintain the consistency of the neural network between train and test). We used the SGD optimizer with learning rate of .05 with cosine decay over 20000 steps, .9 momentum and batch size of 250.

For Figure 6a, we used the above setup, but with a learning rate of 0.01 and a much higher batch size of 2000, so as to replicate the edge of stability phenomenon [7] which only occurs at high batch sizes. For Figure 6b, we used a learning rate of 0.3 and batch size of 32, so as to show the behavior of high learning rate and small batch size on train loss.

### A.2.2 CIFAR-10 Multiple Passes

In Figure 10, we show the dynamics and representational consistency of ResNets trained on CIFAR-10 for several epochs. The architecture is a ResNet-18 with base-shape width set at $N = 64$ channels. The model is trained with SGD with learning rate 0.1 and cosine annealing schedule. The batch-size used is 128.

### A.2.3 ImageNet

In all ImageNet experiments, we used a training subset of the ImageNet-1k dataset consisting of $2^{20} = 1048576$ labeled images and a test subset consisting of $1024$ labeled images. Both subsets were randomly sampled from the full ImageNet-1k training and validation datasets, respectively. To extend the duration in training in which the network remains in the online regime beyond one epoch, we heavily augmented the images in the training dataset using PyTorch's `AutoAugment` transform with the default policy, `AutoAugmentPolicy.IMAGENET`.

We again used the ResNet-18 architecture with $\mu$P parameterization relative to the ResNet-18 network with base-shape width $N = 64$ channel [13]. All architectures and training procedures were implemented in Jax and used the auxiliary Flax and Optax packages, respectively.

Figures 2(b) and 7(b) were trained using the Adam optimizer with the following learning rate schedule: linear warm-up for 0.5 epochs from learning rate $8 \times 10^{-5}$ to $8 \times 10^{-3}$, followed by cosine decay over $49.5$ epochs to $8 \times 10^{-5}$.

## A.3 Language

### A.3.1 Wikitext-103 Language Modeling

For all Wikitext-103 tasks, we adopted the $\mu$P transformer as defined in the $\mu$P package [13]. In the plots shown in the main text, we used a depth-4 transformer, with $d_{model}, d_k, d_v = N$ and $d_{ffn} = 4N$. We performed a single pass through the train set in order to stay in the realistic online regime. We used a masked language modeling with sequence length $S$ at varying input sequence lengths $S$. For Figure 1 d) we used the $S \times S$ attention matrix of an $S = 128$ transformer. In Figure 5 e) we used the attention matrix of an $S = 35$ transformer. We chose this different length simply to

illustrate the consistent message across sequence lengths. We used a batch size of $B = 32$ for all experiments. The residual stream was thus a tensor of shape $(S, B, d_{model})$.

We used the Adam optimizer with a learning rate of $0.0001$. We also ran the same configuration with SGD and a learning rate of $0.5$ and observed the same behavior. See section B for further plots and details.

For figure 3, we used the Wikitext-103 validation set in order to measure the evolution of the predictions on masked logits. In 3 f), we averaged the mean squared error from the widest transformer by using 100 test points.

### A.3.2    C4 Language Modelling

Figure 1 (b) we trained with base network being a 125m parameter transformer model on 2.5 billion tokens using the Mosaic ML's LLM codebase (`https://web.archive.org/web/20230519184343/https://github.com/mosaicml/examples/tree/main/examples/llm`). See `https://web.archive.org/web/20230519183813/https://github.com/mosaicml/examples/blob/main/examples/llm/yamls/mosaic_gpt/125m.yaml` for the full hyperparameter details. We were limited by time and computational resources in our ability to explore further details of the C4 transformer model.

## B    Further Plots of Convergence

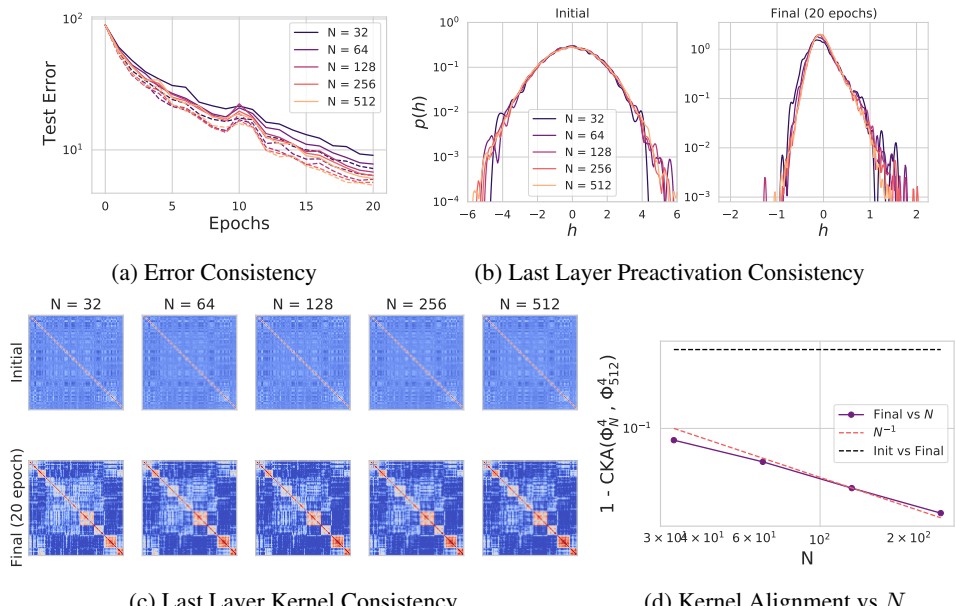

(a) Error Consistency

(b) Last Layer Preactivation Consistency

(c) Last Layer Kernel Consistency

(d) Kernel Alignment vs $N$

Figure 10: Training on CIFAR-10 in a ResNet-18 for multiple epochs generates dynamic preactivation densities and feature kernels which converge at realistic widths. (a) The test classification error curves for single models (solid) and ensembled (dashed) converge for realistic widths. (b) At initialization preactivation distributions in the last hidden layer of the CNN for a randomly chosen data point are Gaussian (as expected) and are very consistent across model widths $N$. To obtain histograms we train an ensemble of $E = 8$ independently initialized networks concatenate activation patterns across members of the ensemble. After 20 epochs of training (models are around $\sim 95\%$ accuracy), the preactivation distributions for the same data point have become non-Gaussian (consistent with infinite width theory) but are still remarkably consistent for large widths. (c) The final layer's feature kernel at initialization shows very little structure, but (d) after training networks of all widths converge to similar kernels. The plot in (d) compares ensemble averaged kernels with the $N = 512$ ensembled kernel.

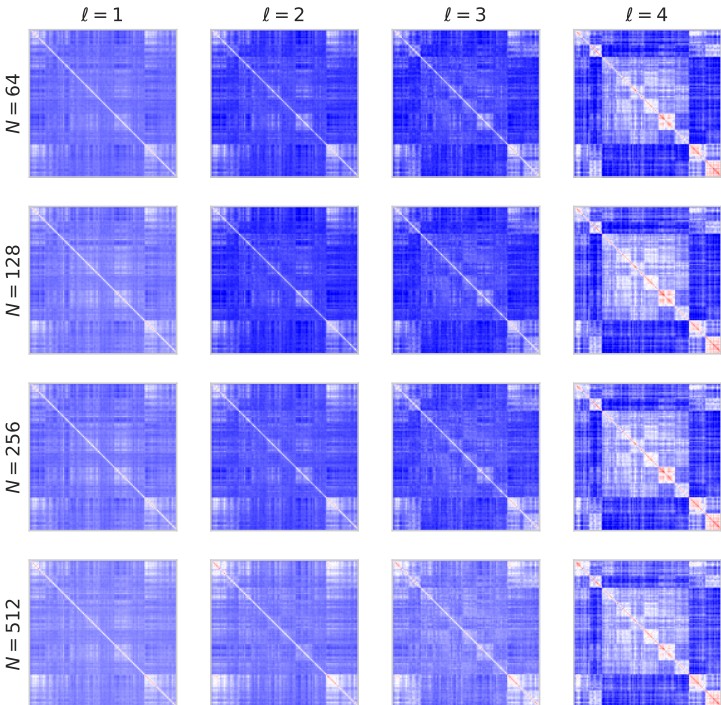

Figure 11: Convergence of layerwise representations in each layer (block) $\ell$ of the ResNet-18 at large width $N$ after training on CIFAR-5M.

In this section, we show additional figures illustrating convergence of network quantities across widths that we did not have space for in the main text.

## B.1 Vision

A simple setting in which convergence properties are particularly clear and simple to study is for a ResNet learning CIFAR-10 and going over multiple passes of the dataset. In Figure 10 we plot a 20-epoch pass over CIFAR 10, and study the generalization error, initial and final preactivations in the last layer, and final layer kernels across widths. The training error begins to exhibit pathologies after sufficiently many epochs, related to the discussion in section 3.2.

Next in figure 11, we show a higher-resolution plot of the kernel Gram matrices across widths and across layers for the CIFAR-5M ResNet after a pass through the data. The larger resolution allows one to see that even the fine-grained details in the structure of the Gram matrix are consistent across widths.

## B.2 Language

Next, in Figure 12, we create an analog of the language column of Figure 3, this time for $\mu$P transformers of the same architecture and dataset but now optimized with vanilla SGD. The fact that wider transformers perform better still holds, and one can clearly see narrower networks approaching wider ones in their output logit values.

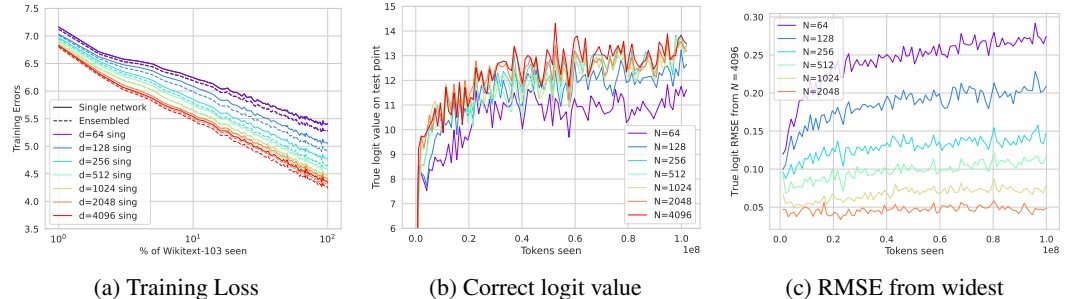

(a) Training Loss      (b) Correct logit value      (c) RMSE from widest

Figure 12: An analog of Figure 3 for $\mu$P transformers trained with SGD. a) Training loss. It is interesting that in $\mu$ parameterization the SGD optimized network is competitive with the Adam-optimized network. b) Value placed on the correct logit for a specific masked token. c) RMSE of correct logit value from the widest network.

## C    Defining $\mu$P and SP (Standard Parameterization)

There are several detailed discussions about $\mu$P vs SP scaling [12, 15, 9, 10, 14]. The aim of this section is to simply give an accessible and conceptual overview of their distinction, as well as a motivation for $\mu$P from the perspective of keeping features moving in time even at infinite width. .

There are several equivalent ways of parameterizing neural networks that give rise to the same dynamical effects, whether in $\mu$-parameterization or standard parameterization. We give the definitions in the case of a single-output feed-forward network and demonstrate that SP and $\mu$P give rise to $\Theta(N^{-1/2}), \Theta(1)$ feature movement at initialization, respectively.

Generalizations to other architectures (ResNets, Transformers) are straightforward. For a detailed discussion see [12] and also [14].

### C.1    SP

We assume all hidden layers have equal width $N$. Let the input space have dimension $D$. Let $\mu$ be the index of the training point in the dataset. At each layer $\ell$, the pre-activation $\boldsymbol{h}_\mu^{\ell+1}$ in layer $\ell + 1$ is given by

$$\boldsymbol{h}_\mu^{\ell+1} = \frac{1}{\sqrt{N}} \boldsymbol{W}^\ell \cdot \phi(\boldsymbol{h}_\mu^\ell), \tag{2}$$

where $\phi$ is an element-wise non-linearity, often taken to be the ReLU function. Here the $N^{-1/2}$ out front allows $\boldsymbol{h}_\mu^{\ell+1}$ to be $\Theta(1)$ at initialization as $N \to \infty$ by the law of large numbers. The output of the network $f_\mu$ is then given by:

$$f_\mu = \frac{\alpha}{\sqrt{N}} \boldsymbol{w}^L \cdot \phi(\boldsymbol{h}_\mu^\ell). \tag{3}$$

Here again the $N^{-1/2}$ scaling again yields that $f_\mu$ will be $\Theta(1)$ as $N \to \infty$. In SP, $\alpha$ is taken to be 1, but we will keep it explicit as it plays an important role in distinguishing the parameterizations. It is the laziness parameter identified in [31]. The change in the function is given by

$$\frac{df_\mu}{dt} = -\eta \sum_\nu K_{\mu\nu} \ell'(f_\nu, y_\nu). \tag{4}$$

Here $K_{\mu\nu} = \nabla_\theta f_\mu \cdot \nabla_\theta f_\nu$ is the NTK gram matrix. $y^\nu$ is the true label. $\ell$ is the loss function (e.g. MSE or crossentropy) and $\ell'$ is its derivative with respect to the first argument. The NTK is easily seen to be order $\alpha^2$ and $\ell'$ is order 1 at small $\alpha$. In order to have the change in the function be $\Theta(1)$ we set $\eta = \eta_0/\alpha^2$.

Using the chain rule, one can directly see that the pre-activations evolve as [10, 3, 32]

$$\frac{d\boldsymbol{h}^\ell}{dt} \sim \eta \frac{\alpha}{\sqrt{N}} = \frac{\eta_0}{\alpha\sqrt{N}}. \tag{5}$$

Thus, at large $N$ and $\alpha = 1$ the pre-activations of this network evolve as $\Theta(N^{-1/2})$. Consequently, at infinite width the feature do not evolve and infinitely wide networks in standard parameterization become kernel machines with the static and initialization-independent infinite-width NTK.

In many machine learning libraries, the factors of $1/\sqrt{N}$ are not explicitly placed in front of each multiplication with the weight matrices. Rather, the weight matrices themselves are drawn from a distribution $\boldsymbol{W}^\ell \sim \mathcal{N}(0, \frac{1}{N}\mathbf{1})$. Although this gives identical forward pass, this changes the $N$ scaling of the gradients in the backward pass. As long as the learning rate is appropriately rescaled to account for this, the dynamics are equivalent to the SP parameterization discussed above.

## C.2 $\mu$P

One of the simplest ways to define the $\mu$-parameterization is to take $\alpha = 1/\sqrt{N}$. This implies that we simply replace the final layer of the network by:

$$f_\mu = \frac{1}{N}\boldsymbol{w}^L\phi(\boldsymbol{h}_\mu^\ell). \tag{6}$$

As the prior analysis shows, in order to have $df_\mu/dt$ be $\Theta(1)$ at initialization, we take $\eta = N\eta_0$, so the learning rate in this definition scales extensively with $N$. In this setting, we now have that

$$\frac{d\boldsymbol{h}^\ell}{dt} \sim \Theta(1). \tag{7}$$

In [12, 15], gives an equivalent definition of $\mu$P that gives rise to the same dynamics but keeps the learning rate to be $\Theta(1)$. We use this version of $\mu$P in the experiments that we run, simply because that is what is used in the package [13]. Consequently, our learning rate does not need to be changed as width grows.

# D   Importance of Parameterization

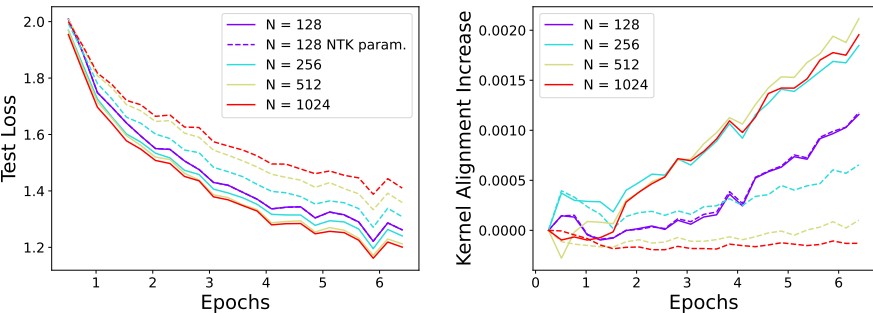

Figure 13: (a-b) An experiment on CIFAR-10 with a depth $4$ CNN (no layernorm) showing that the finite width consistency at the range of widths $N$ considered in this work is special for $\mu$P and is not observed in NTK parameterization. Width-consistent kernel regime behavior requires a much larger width. Networks are parameterized to agree at $N = 128$. We see that in NTK parameterization wider networks train more slowly and their kernels align less to the target function. The reason for the stronger consistency of $\mu$P networks is that feature learning remains $\Theta(1)$.

In this section, we aim to illustrate that many of the claims of width consistency at practical widths are in fact contingent on using the mean field/$\mu$-parameterization. In Figure 13 we compare the early dynamics of network training on CIFAR-10 between networks in the NTK parameterization (dashed lines) and in mean field parameterization (solid). The models are initialized in such a way so that their $N = 128$ dynamics perfectly coincide. We plot both test error and the kernel alignment between the NTK and the target function. While both parameterizations will eventually converge to an infinite width limit, we note the following two differences in width scaling behavior:

1. Wider networks tend to train more slowly in NTK parameterization and perform worse as the width grows. This is because the rate of feature learning is not held constant across widths and decreases as width grows. Wider networks in $\mu$P train slightly faster.

2. Mean field networks approach their limit more quickly as width increases in terms of both test loss and kernel dynamics.

The general theoretical principle behind these observations is that mean field parameterization generates feature updates which are scale independent, thus eliminating an unnecessary source of finite size approximation error in the dynamics [39].

# E   Further Studies of $\mu$P versus SP

## E.1   MLPs

In figure 14, we show a 3-layer MLP learning a quadratic polynomial. In subfigure a) use a batch size of 10 and a learning rate going as $\eta = 5N/(1 + \alpha_0^2)$ with $\alpha_0 = 1$. The output layer is scaled as $\alpha_0/\sqrt{N}$, putting us in the rich regime. The learning rate has been picked to be nearly as large as possible at this batch size in order to maximize the large loss curve fluctuations yielded by large learning rate effects. In subfigure c) we do not rescale the output layer, and have a width-indepdent learning rate going as $\eta = 50/(1 + \alpha_0^2)$ with $\alpha_0 = 1$. This puts is in the large-learning rate regime for a standard parameterized network. See Appendix A for more details.

We plot the learning curves across widths and find striking agreement, even at the fine-grained level of fluctuations due to small batch size and large learning rate effects. Although this is not exactly the full-batch edge-of-stability effect reported in [7], the large oscillations may be similar to a small batch size analog. We plot the absolute difference from the widest network in subfigure b) to highlight the strong agreement across widths.

In subfigure c), we have the same network but in standard parameterization. The narrower networks now learn features more quickly, leading to inconsistent dynamics across widths.

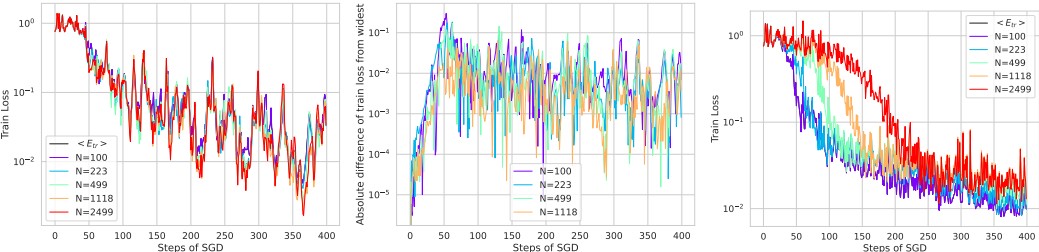

Figure 14: 3-Layer MLP learning a quadratic polynomial $y = Q_2(\boldsymbol{\beta} \cdot \boldsymbol{x})$ on the sphere. Data is provided in an online setting in the same order across widths, as in the realistic experiments in the main text. Large learning rate small batch size effects in MLPs are consistent across large widths. a) For $\mu$P networks, the loss curves match across widths, even accounting for fluctuations due to batch size or large learning rate edge-of-stability-like effects. b) Plot of the difference in training error from the widest network. c) The same network but in standard parameterization. Dynamics are no longer consistent across widths, and wider networks approach a lazy limit.

## E.2   Vision

Next, we focus on a vision task and compare the large learning rate small batch size effects in SP to the $\mu$P parameterized network in Figure 6a. By contrast to that figure, we see significantly different dynamics and batch variation across widths. In Figure 15 a) we plot the early time behavior of a CIFAR-5m task at large learning rate. The large learning rate effects cause the loss to substantially oscillate, but the oscillations across widths are inconsistent by contrast to 6a b) . Further, at late times in Figure 15, the sharp spikes in the loss function due to large learning rate effects become substantially different across widths. Indeed, in SP some widths converge for a given learning rate while others do not. This trend has already been well-studied in [15]. We again stress that our observation is that not only are the final losses similar across widths in $\mu$P (as observed in [15]), but that the individual batch and large learning rate fluctuations agree across widths at early times in $\mu$P as well.

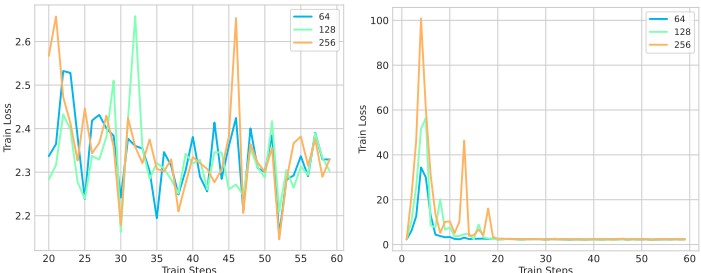

Figure 15: Different widths have different loss curves. a) Early time dynamics of the loss across widths is not consistent. b) Dynamics of the loss across widths at later times also does not appear consistent. There are explosions that happen at different times and scales across widths.

## E.3 Language

Finally, we present a complementary set of figures to those in the right columns of Figures 3 and 12 for transformers of the same architecture on Wikitext-103 but in standard parameterization.

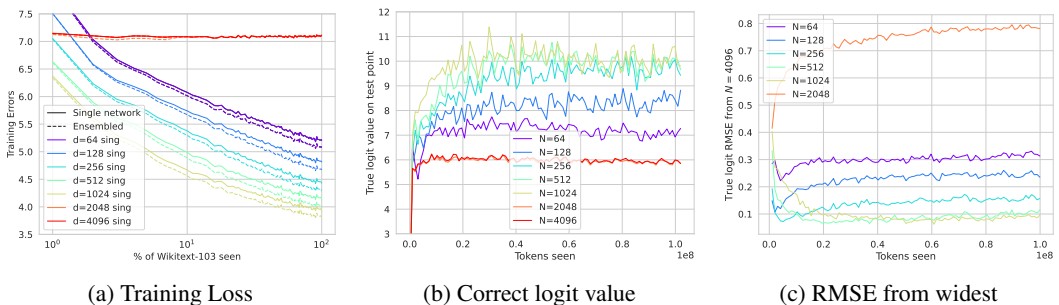

(a) Training Loss      (b) Correct logit value      (c) RMSE from widest

Figure 16: An analog of Figure 3: SP transformers trained with Adam. a) Training loss. At large widths, the learning rate chosen is too big for the network to properly learn, and the loss is flat. This is consistent with what is observed in [15] — the optimal learning rate in SP changes with width. b) Value placed on the correct logit for a specific masked token. c) RMSE of correct logit value from the widest network. In both of these plots, the monotonic behavior across width evident under the $\mu$P parameterization is violated. Even after discarding the networks that do not converge under SP, the behavior remains non-monotonic across width.

## F  Bias-Variance Decompositions over Initializations

In this section we explicitly define what we mean by initilization bias and initialization variance. Following [30], we consider the trained neural network function $f_{\theta^*}(\boldsymbol{x})$ to depend on the training set $\mathcal{D}$ (including inputs $\boldsymbol{x}$, outputs $y$, and possible label noise $\epsilon$) as well as the initial parameters $\theta_0$. Here, $\theta_0$ are the initial parameters and $\theta^*$ are the final parameters, which are implicitly functions of the initial ones.

Classical statistical learning theory often focuses on the variance of the learned function as a function of the training samples given $(\boldsymbol{x}_\mu, y_\mu)_{\mu=1}^P$. By contrast, our paper focuses on the variance due to the initial parameters $\theta_0$ from which training begins. In the overparameterized regime with more parameters than data-points and relatively little label noise, this has been shown to be the dominant source of variance for neural networks [27, 29, 28].

Consequently, our definition of the *bias* of a neural network is given by

$$\overline{f}(\boldsymbol{x}) = \mathbb{E}_{\theta_0}[f_{\theta^*(\theta_0)}(\boldsymbol{x})]. \tag{8}$$

The bias can be approximated by averaging a sufficiently large ensemble of neural networks over initialization seeds. Each network in the ensemble is trained on the same dataset in the same batch order using the same optimizer.

The *variance* of the neural network predictor is given by:

$$\text{Var}_{\theta_0} f = \mathbb{E}_{\theta_0}[(f_{\theta^*(\theta_0)}(\boldsymbol{x}) - \bar{f}(\boldsymbol{x}))^2]. \tag{9}$$

The variance of $E$ ensembles of a given network with independent initialization seeds $\theta_e$ is given by

$$\text{Var}_{\theta_0}\left[\frac{1}{E}\sum_e f_{\theta^*(\theta_e)}(\boldsymbol{x})\right] = \mathbb{E}_{\theta_0}\left[\left(\frac{1}{E}\sum_e f_{\theta^*(\theta_e)}(\boldsymbol{x}) - \bar{f}(\boldsymbol{x})\right)^2\right]. \tag{10}$$

As $E \to \infty$, the empirical average of $E$ network ensembles approaches the bias, so the variance of the ensembled networks goes to zero. As long as the errors are uncorrelated, the variance the the network ensembles decreases as $1/E$. In practice, we see that comparable decay rates are achieved.

## G   Task-Dependent Scaling Laws in Width

For the settings where we observed larger deviations in the dynamics for models of varying widths, we examined scaling of the training losses with respect to width $N$ after a significant amount of training (2 epochs for ImageNet and 1 epoch for Wikitext 103). We fit power laws of the form Loss $= CN^{-\beta} + D$ where $C, \beta, D$ are fit to the data using the 'scipy.optimize' function. The resulting fits are provided in Figure 17. We find an excellent power law fit, with $R^2$ above .99.

The existence of such power law behavior across widths provides further evidence of the networks approaching a well-defined inifinite width limit. One could imagine that the differences between successive networks might get smaller (as in Figure 4) but that there is no well-defined limit as $N \to \infty$, similar to the terms in a harmonic sum. The fact that the power law fit has exponent much larger than 0 and does not display logarithmic dependence on $N$ provides empirical evidence that we expect convergene as $N \to \infty$.

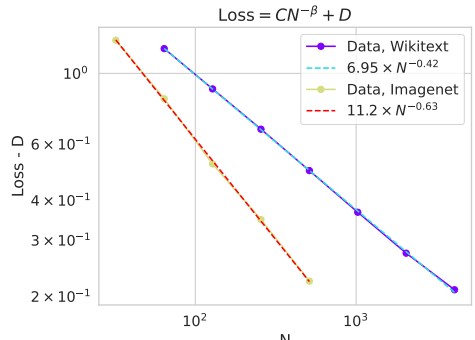

Figure 17: Train losses of Wikitext and Imagenet models are well-fit by power laws.

Additionally, given that the observed power laws $N^{-\beta}$ are task-dependent and significantly different from $N^{-1}$ suggests that models at late time are not well described by perturbation theory around the infinite width mean field limit, which predicts a universal exponent of $\beta = 1$ [54, 39]. This motivates novel theoretical descriptions of finite width mean field learning networks at late time which can capture task-dependent exponents.

## H   Overview of Finite Width Corrections to Feature Learning Networks

In this section we review some basic ideas from the mean field theory of feature learning neural networks. We first describe the predictions that mean field theory makes about infinite width networks before describing finite size corrections to the dynamics of learning. To eliminate unnecessary complexity, we will focus on MLP layers, but these arguments can be easily extended to CNN and self-attention layers as well. We start by defining a MLP in a parameterization equivalent to $\mu$P

$$f_\mu = \frac{1}{N}\boldsymbol{w}^L \cdot \phi(\boldsymbol{h}_\mu^\ell)\,,\; \boldsymbol{h}_\mu^{\ell+1} = \frac{1}{\sqrt{N}}\boldsymbol{W}^\ell \phi(\boldsymbol{h}_\mu^\ell)\,,\; \boldsymbol{h}_\mu^1 = \frac{1}{\sqrt{D}}\boldsymbol{W}^0 \boldsymbol{x}_\mu. \tag{11}$$

We will consider these networks trained from a random Gaussian initialization of the weights so that $\boldsymbol{\theta} = \text{Vec}\{\boldsymbol{w}^L, ..., \boldsymbol{W}^0\}$ follows $\boldsymbol{\theta} \sim \mathcal{N}(0, \boldsymbol{I})$ at initialization. This network is then trained with some gradient based optimizer, leading to dynamical predictions $f_\mu(t)$ and dynamical preactivations $\boldsymbol{h}_\mu^\ell(t)$. Because of the random initialization of weights, the outputs of the network and the precise preactivations are random variables. However, at infinite width $N \to \infty$, a dramatic simplification of the dynamics occurs.

## H.1 The Infinite Width/Mean Field Limit

The predictions $f_\mu(t)$ and internal representations of infinite width limit of neural networks admit a description in terms of non-random initialization-independent dynamical feature kernels $\Phi^\ell_{\mu\nu}(t,s)$ and gradient kernels $G^\ell_{\mu\nu}(t,s)$ defined as

$$\Phi^\ell_{\mu\nu}(t,s) = \frac{1}{N}\phi(\boldsymbol{h}^\ell_\mu(t)) \cdot \phi(\boldsymbol{h}^\ell_\nu(s)) \,, \ G^\ell_{\mu\nu}(t,s) = \frac{1}{N}\boldsymbol{g}^\ell_\mu(t) \cdot \boldsymbol{g}^\ell_\nu(s), \tag{12}$$

where $\mu, \nu$ index data points and $t, s$ index training time and $\boldsymbol{g}^\ell_\mu(t) = N\frac{\partial f_\mu}{\partial \boldsymbol{h}^\ell}$ are back-propagated gradient signals [12, 14]. Further, all preactivation vectors $\boldsymbol{h}^\ell_\mu(t) \in \mathbb{R}^N$ have entries that become iid draws from a (potentially non-Gaussian) single site density $p(h)$, which converges as

$$\frac{1}{N}\sum_{i=1}^N \delta(h - h_i) \to p(h), \tag{13}$$

which should be understood in terms of integration of these densities against test functions. At infinite width, the sums over neurons in a layer can be replaced by deterministic integrals over this single site density $\Phi^\ell_{\mu\nu}(t,s) = \int p(h^\ell_\mu(t), h^\ell_\nu(s))\phi(h^\ell_\mu(t))\phi(h^\ell_\nu(s))dh^\ell_\mu(t)dh^\ell_\nu(s)$.

## H.2 Finite Width Effects

At finite width, the internal kernels $\{\Phi^\ell_{\mu\nu}(t,s), G^\ell_{\mu\nu}(t,s)\}$ and predictions $f_\mu(t)$ of the model become initialization and width-dependent and deviate from their mean field dynamics. For Gaussian random initialization of the weights of the network, the predictions and kernels fluctuate (from init to init) with variance that scales asymptotically like $\Theta(1/N)$ for width $N$ (or $1/d_{model}$ for transformer) [39]. Further, the *ensemble averaged* values for the predictions $\langle f_\mu(t)\rangle$ and kernels $\langle \Phi^\ell_{\mu\nu}(t,s)\rangle$ differ asymptotically from their infinite width values by $\Theta(N^{-1})$. Both of these two leading order effects can influence the expected (train or test) loss of the model. At fixed width and late training time, finite size effects beyond leading order can accumulate and become relevant, however theory predicts that any observable average at width $N$ admits an asymptotic series in powers of $N^{-1}$ [39].

### H.2.1 Trainability at Finite Size

The $\Theta(N^{-1})$ correction to feature and gradient kernels can lead to non-trivial corrections to the loss dynamics. Working in continuous time, we can define the neural tangent kernel (NTK) as $K_{\mu\nu}(t) = \sum_\ell G^{\ell+1}_{\mu\nu}(t,t)\Phi^\ell_{\mu\nu}(t,t)$, where base cases are $\Phi^0_{\mu\nu}(t,s) = \frac{1}{D}\boldsymbol{x}_\mu \cdot \boldsymbol{x}_\nu$ and $G^{L+1}_{\mu\nu}(t,s) = 1$. Following the approximation to online dynamics with MSE loss in Section 4, we consider a gradient flow on the average dynamical NTK

$$\frac{d}{dt}\boldsymbol{\Delta}(t) = -\langle \boldsymbol{K}(t)\rangle \boldsymbol{\Delta}(t) \implies \boldsymbol{\Delta}(t) = \mathcal{T}\exp\left(-\int_0^t ds\, \langle \boldsymbol{K}(s)\rangle\right)\boldsymbol{y}, \tag{14}$$

where $\mathcal{T}$ is the time-ordering operator. We now consider the leading correction to the average NTK around infinite width $\langle \boldsymbol{K}(t)\rangle = \boldsymbol{K}_\infty(t) + \frac{1}{N}\boldsymbol{K}^1(t) + \Theta(N^{-2})$. With this correction, we see that the dynamics of errors $\boldsymbol{\Delta}$

$$\boldsymbol{\Delta}(t) = \mathcal{T}\exp\left(-\int_0^t ds\, \boldsymbol{K}_\infty(s) - \frac{1}{N}\int_0^t ds\, \boldsymbol{K}^1(s) + \Theta(N^{-2})\right)\boldsymbol{y}. \tag{15}$$

The fact that the $\frac{1}{N}\boldsymbol{K}^1$ correction is integrated over time and placed in the matrix exponential indicates that small corrections to NTK dynamics can lead to large dynamical amplification of logit corrections. This fact was pointed out in another work [39] which tried to motivate a study of perturbation theory in logarithms of the transition matrix $\log \boldsymbol{T}(t)$ defined as

$$\frac{d}{dt}\boldsymbol{T}(t) = -\langle \boldsymbol{K}(t)\rangle \boldsymbol{T}(t) \,, \ \boldsymbol{T}(0) = \boldsymbol{I} \,, \ \boldsymbol{R}(t) = \log \boldsymbol{T}(t). \tag{16}$$

The solution to this can be used to construct the errors at a later time $\boldsymbol{\Delta}(t) = \exp\left(\boldsymbol{R}(t)\right)\boldsymbol{y}$.

# I The empirical sufficiency of ensembling only a few times

Our experiments rely on ensembles of small numbers of neural networks to analyze the bias component of the loss as it varies across width. One can show the marginal value of adding a network to the ensemble decreases with the ensemble size. Figure 18 illustrates this in the setting of ResNets trained on ImageNet. In Figure 18(a), we show that the reduction in variance over initializations due to ensembling rapidly plateaus as soon as the ensemble size reaches $E = 3$. In Figure 18(b), we show that the loss curves as function of width are very similar for ensemble sizes above 3. Indeed, the green, orange, and red curves — corresponding to $E \in \{3, 4, 5\}$ — are nearly identical.

Lastly, Figure 18(c) confirms that the initialization variance plays a negligible role in the scaled RMSE distances between true logits across width, even for single draws of networks trained on a small number of examples. The dashed lines in this Figure correspond to scaled RMSE distances between networks of the same widths as their solid-line companions, albeit with weights initialized to zero. The ordering and scale of the curves are comparable, and in fact nearly identical for the curves corresponding to $N \in \{128, 256\}$ compared to $N = 512$.

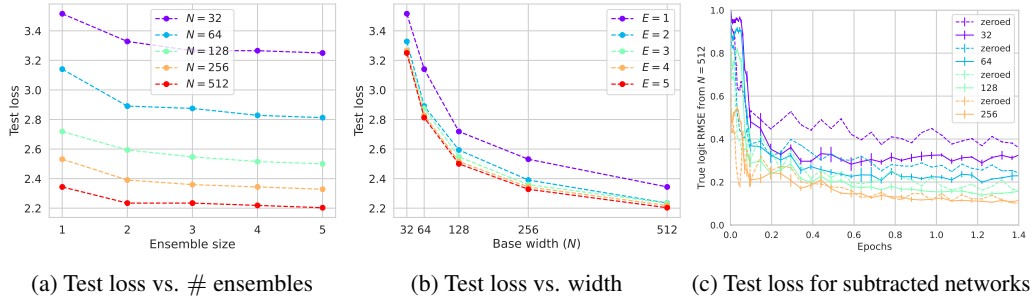

(a) Test loss vs. # ensembles     (b) Test loss vs. width     (c) Test loss for subtracted networks

Figure 18: ResNet-18/ImageNet loss as a function of a) ensemble size or b) width after 2 epochs of training with heavy data augmentation. c) Comparing original network with a network whose output has been set to zero at the start of training.

# J Offline Training

Figure 19 depicts the loss curve for a ConvNeXt-T (tiny) model trained on ImageNet in the typical, offline setting — where data is encountered repeatedly across many epochs. As the networks overfit the training data — in Figure 19, beyond 40,000 training steps or five epochs — the loss curves diverge dramatically for different-width networks. Width consistency subsequently erodes.

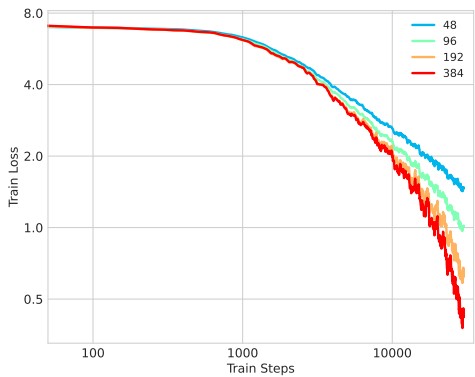

Figure 19: Consistency of loss curves across widths in the beginning and separation as loss becomes sufficiently small in offline learning.

# K   Use of Compute

For most experiments, we used Nvidia A100 SXM4 40GB and 80 GB GPUs on an academic cluster.

For the Wikitext-103 tasks, each width included 4 ensembles loaded onto an A100 GPU that ran for a range between 1 to 3 days. For each sweep over widths this corresponds to about 8 A100-days. Accounting for sweeps over different sequence lengths, optimizers, and parameterizations, this corresponds to about 50 A100-days.

All MLP tasks, including the calculation of empirical NTKs and their spectral properties were done in 15-30 minute Colab sessions using the basic GPUs provided.

The CIFAR-10 ResNet experiments in Figure 10 were done using a total of less than 1 A100-day of compute across all widths and ensembles.

The ImageNet ResNet experiments vectorize training over between one to four same-width neural networks on one A100 GPU. Each experiment training a collection of networks for 30 epochs takes between one to three A100-days. Overall, these experiments expended roughly 30 A100-days.

For the CIFAR-5m experiments in Figure 2 and 3, across all widths, it required a few hours of A100 GPU. For Figure 7 and 8, as these were ensembled across multiple runs, these required close to 1-2 days of A100-GPUs. Figure 6a was just run for a few 100 steps of the training, so didn't use much compute power.

