# OpenReview forum: "Feature-Learning Networks Are Consistent Across Widths At Realistic Scales"
_NeurIPS.cc/2023/Conference — NeurIPS 2023 poster_

### Official Review · Reviewer_j9zd · 2023-06-26

**Soundness:** 2 fair
**Presentation:** 3 good
**Contribution:** 2 fair
**Rating:** 6
**Confidence:** 3

**Summary:**

This paper studies the convergence of key quantities (like test predictions, feature kernels) in Neural Networks as width increases, in feature-learning regimes. The paper is largely an empirical study, and presents width-convergence results on both image and language tasks. The paper also uses ensembling to consider the role of finite-width variance from randomness in initialisation, and finishes with a toy model that looks into the causes behind the bias in finite-width models compared to their infinite width counterparts.

**Update: after clarifications in the rebuttal and additional experiments, I am raising my score to 6**

**Strengths:**

1. The question that this paper seeks to address, i.e. understanding finite-width NNs, is important and open.
2. The paper has some interesting conclusions e.g. the convergence of feature-learnt test predictions at realistic widths (though see weaknesses too for some concerns), or the breakdown of finite-width bias in terms of eigenfunctions rather than eigenvalues.
3. The paper is well-written and clear.

**Weaknesses:**

1. My main concern is that some of the results are not convincing/not matched to the claims of the paper. For example, for the qualitiative results on ImageNet experiments on Figure 2b and 3b, it isn't clear to me that these curves have converged in width (say comparing N=256 or 512). For the quantitative equivalent to 3b, Figure 3e, it's a bit surprising that the initial y-intercepts on the true logits RMSE with width 512 are so large, even at width N=256 (and seemingly not consistent with the claim that these predictions have converged at these evaluated widths). This is especially confusing given that it seems one of the claims in Figure 2 is that ImageNet results have converged for earlier training times compared to later times, and suggests that the reason why the True logit RMSE to width 512 is small at later epochs is simply because different networks are converging to the same correct prediction, rather than any width-limit occuring. Similar conclusions (larger RMSE at earlier training steps) appear to be the case in Figure 3f for WikiText-103.
2. Likewise, I find the fact that we are often judging convergence in width off of 3-5 widths (e.g. Figure 1a-c) to be a bit unconvincing. I realise computational constraints are likely at play here, but it would be very good for the contribution of the paper to have more widths presented. Moreover, the argument that "a quantity's deviations between successive widths decrease as widths are increased" (line 54) is evidence for consistency is a bit weak to me: the harmonic series blows up to infinity despite satisfying this property.
3. Going back to the motivation (e.g. line 86), the extent to which establishing convergence to infinite-width feature learning limits helps the community in understanding finite-width NNs might be quite modest, as such limits are very difficult to understand (significantly harder to understand compared to say kernel regimes), and also computationally prohibitive (e.g. Yang and Hu 2020, or Bordelon and Pehlevan 2022)

**Questions:**

Please respond to be weaknesses above, and also answer:

1. Do you set the output weights to 0 in order to reduce the variance from function initialisation in your results? If not, then it might be worth looking at e.g. for the ensembling results.
2. Do the experiments use identical mini-batch orderings across different widths?

Also, please correct the following typos:

1. 'we put all of our in...' (line 37)
2. 'the training lost' (figure 2 caption)

**Limitations:**

There isn't much discussion of limitations within the submission.

---

> ### Author Rebuttal · Authors · 2023-08-09
>
> We thank the reviewer for these thoughtful comments. We'd like to address them in sequence.
>
> * *for the qualitiative results on ImageNet experiments on Figure 2b and 3b, it isn't clear to me that these curves have converged in width (say comparing N=256 or 512).*
>
> This is a good point. In the main paper, we have not included enough plots of quantities as functions of $N$. This is something we plan to fix. In Fig 5 of our supplement we have plotted the final loss value as a function of $N$ (as well as ensemble number) for the Imagenet experiment in 2b. In Fig 3, A scipy.curve_fit on our losses as functions of $N$ gives an excellent fit to the power law $C N^{-\beta} + D$.  A similar fit works for the transformer model as well. Moreover, the exponent $\beta$ is frequently found to lie between 0.3 and 1.0. This power law dependence is also present in figures 4c, f, although is more cleanly seen when tracking losses rather than representation differences.
>
> For Fig 3b, it is true that the logits tend to grow over the course of training. In this setting, we could have plotted the change relative  to the logit of the widest network, which would give a curve similar to 3e. For 3e we did in fact divide by the widest network's logit variance to rescale everything and keep things constant in time. We will explicitly note this in the updated revision.
>
> * *"...it's a bit surprising that the initial y-intercepts on the true logits RMSE with width 512 are so large..."*
>
> One important aspect that we will clarify in the main text is that in calculating this RMSE we are dividing by the standard deviation of the widest network's outputs. That is:
> $RMSE = \frac{\sum_i (y_i^n - y_i^{widest})^2 }{\sum_i (y_i^{widest})^2 }$
> where $i$ indexes the test points and $n$ is the width of the network we are comparing to. Near the start of training, the outputs are small, especially in $\mu$P, and this causes an early sensitivity in the curves. Once the network's output has grown, it goes away. This is the primary driver of this effect.
>
> * *I find the fact that we are often judging convergence in width off of 3-5 widths (e.g. Figure 1a-c) to be a bit unconvincing.*
>
> It would indeed be nice to go to still-higher values of $N$ but as you point out, we eventually do become limited by compute. For the transformer model, the width 4096 is quite large. Larger width models cannot easily be loaded onto an 80GB A100. We sweep over 7 widths and 2 orders of magnitude from width 64 to 4096 at good resolution. For the imagenet models, the ResNet architecture has $N$ for the initial number of channels. The final number is $8N$, or 4096 for the width 512 network. This is again near our computational limit, and is realistic for what is used in practice.. The range of widths was largely motivated by matching with the model sizes used in practice, to the extent that we could. Finally, the power law fits we observe match the data to high accuracy across the entire range of widths, providing some evidence that our observed trends can be extrapolated to a reasonable degree.
>
> * *Moreover, the argument that "a quantity's deviations between successive widths decrease as widths are increased" (line 54) is evidence for consistency is a bit weak to me: the harmonic series blows up to infinity despite satisfying this property.*
>
> This is an apt observation, and one that worried us when formulating a definition of consistency. When we refer to consistency we simply state that with high probability a given network observable lies in an increasingly small region as the width grows, using language similar to that defining Cauchy sequences. In practice it is better to do a power law fit on a given network quantity and see if the fit is accurate and decaying. With a finite number of data-points this is likely the best one can do. All of our decays are well-fit by power laws with non-negligible exponent in $N$ (see above comment and Fig 3). A diverging sequence like the partial sums of the harmonic series would exhibit a power law exponent that either is close to zero or gets smaller as more points are added.
>
>
> * *"Going back to the motivation..."*
>
> Indeed, the simulations of mean field limits require a time cubic in the number of train steps to approximate. Even kernel methods are for the most part prohibitively expensive for large-scale datasets seen in practice, but mean field limits are so far less interpretable. We believe that a key contribution of this work is that, by contrast to NTK limits, which frequently break at realistic widths due to both finite-width kernel corrections and NTK changes due to feature learning, the $\mu$P limits can be reached achieved at realistic widths to high accuracy at early training times. Further, there are still nontrivial predictions that can be made for infinite-width mean field networks and their finite width corrections even if simulating them is complicated.
>
> **Questions**
>
> * *"Do you set the output weights to 0 in order to reduce the variance from function initialisation in your results? If not, then it might be worth looking at..."*
>
> We did not originally have the outputs subtracted off. We did however look at this change and observed no major qualitative difference in the results. We have included this result as figure 5c in the supplement and will incorporate it as an appendix figure.
>
> In the lazy limit, this subtraction is particularly important, as the output at initialization is $O(1)$ in the width $N$. For $\mu$P, the output scale goes like $1/N$, and we expect such a subtraction to be less important.
>
> * *Do the experiments use identical mini-batch orderings across different widths?*
>
> Yes, we do use the same mini-batch orderings across all widths. We will revise the text to state this explicitly.
>
> * *"Also, please correct the following typos:..."*
>
> Certainly. We have corrected the typos pointed out in our current draft copy. Thank you for catching them.

---

> > ### Comment · Reviewer_j9zd · 2023-08-16
> > **Thanks**
> >
> > Thanks for the rebuttal and new experiments, as well as for the clarifications. I do think the true logits RMSE plots in (3d-3f) are quite misleading in the original submission, and contradict some of the arguments that the paper is trying to make (i.e. feature learning nets converge in width better at earlier training times). I would potentially suggest something like plotting the non-normalised RMSE in the predicted probabilities (post softmax) instead, which would get rid of this dependence on the norms of the output logits which vary throughout training, at the expense of being invariant to additive scalars in logit space.
> >
> > I think Figure 3 of the attached pdf is more convincing of the arguments of the paper compared to the plots in the submission. I would be interested to see Figure 3 plotted at various points of training to see how the convergence in loss changes as the loss becomes better.
> >
> > I have a further question: in Figure 1 of the attached pdf how do you set the learning rates of the different networks?
> >
> > Thanks!

---

> > > ### Author Response · Authors · 2023-08-17
> > > **Reply to Followup Questions**
> > >
> > > We thank the reviewer for the follow up questions.
> > >
> > > Based on the concern about Figures 3d-3f in the original submission, we are now examining non-nomalized RMSE and alternative metrics which use probabilities rather than MSE between logits directly (TV distance and KL divergence). These metrics do not have the confounding issue of small normalization constant at initialization.
> > >
> > > With these new distance metrics (and with or without zero readout), we continue to find an initial transient decrease in the distance between widths very early in training followed by a more gradual increase in the distances between widths over the course of training as the networks begin to reduce the loss. We will state this explicitly in the main text and add the qualification that the best agreement (in logits/probabilities) between all widths tends to occur after a small number of initial training updates. The source of this effect is interesting and worthy of future investigation and we wonder if it is perhaps related to the transient effects of large learning rate dynamics (edge of stability or catapult like effects).
> > >
> > > We are glad the reviewer is interested in Figure 3 in the attached PDF. We also think more plots of this kind should be included in the paper to help understand scaling laws of networks trained in $\mu$P. We agree that it would also be interesting to see how the scaling laws (with width $N$) change over the course of training time.
> > >
> > > On the last question: each model in the Figure 1 of the attached PDF is using the same learning rate. This will guarantee that both networks approach their respective infinite width limits (kernel limit for NTK and the feature learning infinite width limit for $\mu$P) as the widths become larger. However, all networks have the same initial rate of change to the network predictor $\frac{d}{dt} f$ at the first step of gradient descent. Lastly, note that the networks are parameterized to agree perfectly at width $N=128$ and indeed they have identical loss and kernel alignment dynamics. The difference as widths become larger is that the hidden features/kernels will move at different speeds in NTK vs $\mu$P.

---

> > > > ### Comment · Reviewer_j9zd · 2023-08-18
> > > > **Thanks**
> > > >
> > > > Thanks. With the additional experiments and clarifications, I am raising my score to 6.
> > > >
> > > > Regarding the comments on Figure 3d-3f, in some way it is not surprising that the agreement across widths degrades after initialisation, because we know the logits converge to the same point (0) in MuP at initialisation. I think it is interesting to explore if this disagreement across widths is monotonically increasing with training time (in different learning rate dynamics like EoS or catapult as you suggest) or decreases again towards the end of training. This is as I'd also expect that at the end of training the predictions agree too (at least on training data) assuming that the trained networks have successfully fit the data, because they have the same supervised signal. There are of course questions concerning the capacity of the finite width networks at play here relative to the training task. That is, for 'wide enough' networks we'd expect them to start and end training in agreement, and the question is do they take the 'same' path to get there. This sort of exploration is what I was trying to suggest by plotting Figure 3 of the rebuttal at different points of training.

---

### Official Review · Reviewer_8Tjm · 2023-07-07

**Soundness:** 3 good
**Presentation:** 2 fair
**Contribution:** 3 good
**Rating:** 6
**Confidence:** 3

**Summary:**

The authors study the so-called “feature learning” parameterization of neural networks that is given by the maximal update parameterization (muP). This is in contrast to the NTK parameterization for infinite-widths, which leads to the “lazy learning” regime which is claimed to not learn features, and also in contrast to the standard parameterization for finite-width networks, which the authors also claim (I believe in the Appendix) leads to no feature learning in the large-width limit. The central question of this paper is whether a finite-width (but sufficiently wide) network in the muP setting captures the dynamics of its infinite wide counterpart (which the authors empirically examine by proxy via the largest width network they are able to train, which I’ll refer to as the infinite-proxy). They consider this question in the online setting, where new data is presented in each batch, and the offline setting, where data is recycled over epochs.

Throughout the paper, the authors study the following metrics: training loss, train / test error, logits with respect to the true class for a test sample, attention matrices, top eigenvalues, centered kernel alignment, preactivation neuron distributions, and kernel spectra.

The central findings of the paper are:
- Sufficiently wide networks begin to show similar loss, error, and true logit trajectories over the course of training, though this degrades over training on more difficult tasks (e.g. ImageNet vs. CIFAR tasks)
- You can do a bias-variance decomposition of the finite-width networks by considering ensembles of predictors, and the paper claims that, empirically, sufficiently wide networks don’t suffer much from bias nor variance but narrow networks can suffer from both and that the bias is actually a larger negative effect than variance
- In offline training, if the networks start to overfit then narrow and wide networks show that bias and variance measured here to not go to 0 (there is a claim that on test the bias and variance goes to 0 but on train it doesn’t, I will defer comments on this to later)

EDIT: Thank you for your thorough response, I continue to be more on the accept side than reject and opt to keep my score.

**Strengths:**

There are a lot of experiments and perspectives here to support the claims the authors are making and I don’t think I’ve seen a finite-width study of muP yet, myself. Experiments are also done with MLPs, ResNets, and Transformers, which is an extensive enough set of model families to convince readers that these phenomena are not unique to any one model class. I think the conception of each experiment is nice, and it is clear from the explanation of each plot what it is trying to convince me of. It is clear to me that after a certain width, the finite-width networks begin to act similarly, though there are still minor differences (more on this later).

The ensemble experiments give rise to a sort of bias-variance decomposition, though I think this is not formally a bias-variance decomposition because the definition of bias here given is with respect to the infinitely-wide version, considered as the largest width network they can train, but I think the infinitely-wide version of the network is not necessarily the optimal predictor (e.g. E[y|x] in a regression setup), unless it is in this setting and I have missed the explanation why? However, the variance aligns to the definition of variance I am familiar with, and in general I still think this is an interesting definition in-and-of-itself, just not perhaps a formal bias-variance decomposition.

I appreciate the thoroughness of everything and overall I find this to be a useful contribution with respect to studying muP networks, something I haven’t really seen before in such a practical setting. I can see through the empirical results that this parameterization does lead to a reasonably stable behavior when the network width is sufficiently large, and it is a useful result for those who want to study and understand the muP dynamics.

On one hand, I think there is enough here that it will be of use to some people in the community and is worth accepting. On the other hand, I have many comments and questions and I think a second draft and resubmitting to another conference would make this much stronger. I am giving it a weak accept because everything is technically sound and interesting, but I hope the authors consider my comments below.

**Weaknesses:**

I think it could be misleading to say that you are studying feature-learning networks in the sense that I believe standard parameterization also does lead to “feature learning”, though admittedly I don’t think feature learning itself is a well-defined term. I understand that muP is meant to indicate some non-trivial change in the feature kernel, and in the appendix you argue that standard-parameterization leads to no changes in the feature kernel when width goes -> infinity, doesn’t this still leave room for feature learning in SP in the finite-width (in particular, the narrow-width) regime? I’m not totally sure how you could assess at what point a SP model is doing feature learning or not, but I am open to hearing thoughts on this. Basically, if the point of the paper is to study realism, like realistic scales and realistic models + data then I think it’s too strong to assume that there is no feature learning in realistic SP models. As such, I think it could be more precise to say that you are studying muP models at realistic (finite) scales, and that there are potentially feature-learning dynamics that are not covered in this paper nor encompassed by the muP parameterization. Though I admit I don’t fully know all of the details of muP, so maybe this is just my lack of understanding.

To be more clear, I think it is obvious once you read the paper that you are only studying muP models, so my prior comment is mostly about the title and abstract, which does not refer to muP and could make a potential reader believe that you are studying, somehow, all feature-learning networks. Maybe the paper could make this point about all feature-learning networks if you compared these muP plots to the same for a modern SP model over a few different widths, perhaps a model that is reported from some recent paper on these datasets (they are very well studied datasets so there will be lots of options to choose from). Is muP outperforming a state-of-the-art SP model? If not, can you claim something about whether there is feature learning or not in that SP model? I understand these questions are not the central focus of your paper, and so they are not affecting my decisions on accept / reject. It is mostly a comment that when the title and abstract says you are studying feature-learning networks this is what I immediately think about and it is something I am wondering as I read your paper, and perhaps having that thought in my head undercuts the value of your paper, to me, while I read it.

I think all of the figures are far too small, especially the fonts. It is hard to read without zooming in on the computer, and on printing it is too small. Especially considering that you want to convince the reader of “consistent” behaviors over increasing width and we are meant to observe this by seeing that loss curves, etc become closer on the higher widths, larger and clearer pictures seem very important to me. I think the small figures undersell the evidence of the claim a bit, it’s not always clear to me that some curves are closer while others are not. In that sense, Figure 3(d-f) gives me the clearest evidence of the claims being true, while many of the rest I really have to stare for a bit and zoom in to see the curves more clearly and convince myself that it’s true.

I also feel that there is so much going on here, it’s not necessarily a bad thing but it’s a bit overwhelming as a paper. The sections just seem like they’re going so quickly through results just stating this figure shows this, now this other figure shows something else, etc etc. I feel that I’m missing intuitions, explanations, contextualization in existing literature and more. If I were deeply involved in muP research then maybe all I need is a clearly stated set of results and a wide variety of experiments to cross-reference, but I think the paper could benefit from more exposition.

Please see questions below.

**Questions:**

How small of a difference on a metric like cross-entropy loss is needed in order to consider it a negligible difference, or “consistent” as defined in this paper? I imagine this is specific to dataset, model, and other choices. It would help if you gave some idea of this in the paper.

You could compare the infinite-proxy network (the largest width network that is meant to be a proxy for the infinitely-wide network) to the expected regressor in a simple, synthetic setting (like mixture of gaussians or something like this) in order to see whether the bias of narrow width is like the true bias or it is some new bias-like term you’re proposing, but not exactly equivalent to the actual bias of the model.

On individual model experiments and true logit experiments, are you averaging errors and losses over multiple different initializations when you plot? The ensemble experiment tells me something about averaging predictions and then computing error / loss, but I think since there is such a high variance with respect to random initialization of the network that the losses, and true logit RMSE and other measurements in most experiments should all be reported as an average over initializations (or test points when applicable), and I just haven’t seen anywhere in the paper or appendix that says that this is what you’re doing, though I may have missed it.

I’m a bit confused about the claim in Section 3.2 that on train the bias/variance don’t go to zero and on test they do so this would indicate benign overfitting. I think it may come from the definition of bias-variance you have given, potentially not aligning to the notion of statistical consistency that would indicate benign overfitting. Could you talk a bit about why you think this is showing benign overfitting, and whether your bias-variance is actually the same as statistical consistency?

In the first sentence of the second paragraph of Section 4, starting “Concretely, we see that although the eigenvalue spectrum of the …” I find it quite confusing to read, and I can’t help but feel there must be simpler language to get across the point you’re trying to make. Also what is “the task”? Is it fitting the data? This section reads a bit disconnected to me from the rest of the paper and introduces a lot of new terminology and ideas. While I follow the general point, it feels like you’re bringing in a whole second paper’s worth of topics and then trying to compress it all into one page, which makes it feel very rushed and somewhat out of place to me with respect to the prior experiments. I could be wrong, and maybe my fellow reviewers will disagree with my view on this.

I think you’re arguing that the eigenfunctions and how the weight is placed on them is affected by narrow width more significantly than the spectrum itself, and that this would contribute to your notion of bias being large? To me, it feels that you’re throwing a lot of new ideas at the reader in the end of your paper, like “spread out target function power into slower modes” and “putting more of the task into smaller eigenmodes that take longer to be learned” and none of these ideas are motivated or explained anywhere in the paper? Maybe you are assuming that the reader is very familiar with what these phrases would mean but I’ve read a number of papers in this domain and while I can probably make some assumptions about what you’re trying to say I think it’s a lot to ask of the reader to just go with it, unless they are deeply familiar with this specific vernacular.

---

> ### Author Rebuttal · Authors · 2023-08-09
>
> We would like to thank the reviewer for their detailed review. We address the concerns and questions below.
>
> 1. **Usage of 'bias' does not correspond to usual statistical notion of bias'**: We agree that the way we use bias is not the statistical notion of bias for exactly the reason pointed out by the reviewer: infinitely-wide version of the network is not necessarily the optimal predictor. We will clarify this. We do note that it does correspond to the bias if we take the target function to be the output of the infinitely-wide $\mu$P network. That is, for the MSE between $f_N$ and $f_\infty$, we have $\left< (f_N-f_\infty)^2 \right> =  (\left< f_N \right> - f_\infty)^2   + \left<(f_N - \left< f_N \right>)^2 \right>$.
> 2. **'Feature learning in finite width SP networks'**: We completely agree that finite width SP networks display feature learning. Our aim was to argue that because infinite width SP networks do not have feature learning, infinite width SP networks cannot explain the dynamics of the finite width networks we train in practice which do exhibit feature learning. Therefore SP networks can not be consistent across all large widths. We will rewrite to make this clear.
> 3. **'Usage of mup networks instead of SP networks'**: This is not true since some of the networks we look at are indeed SP network. Note that $\mu$P should not be thought of as a different type of network than finite width standard/NTK parameterization networks, but rather as a rule for how to make a finite width network wider without sacrificing feature learning (preventing a limiting kernel behavior or unstable dynamics). Each $\mu$P network is defined by a base network and a rescaling of the width. If the rescaling factor is set to 1 we recover the original network. We take our base networks to be SP networks (Appendix A.2), hence a) width 64 networks for the CIFAR-5m and ImageNet and b) width 256 for Wikitext are exactly SP networks. So in fact our results argue that SP networks have similar dynamics (up to the discussed deviations) as their corresponding infinite width $\mu$P analogues. We will add a discussion to clarify this. To see an example showing that other parameterizations do not lead to consistent dynamics, please see Figure 1 in the attached rebuttal PDF which shows that kernel and predictor dynamics change with width in NTK parameterization. The base networks at width $N=128$ agree perfectly, but as the width $N$ is increased the two parameterizations ($\mu$P and NTK) approach different limits.
> 4. **‘How small of a difference on a metric like cross-entropy loss is needed in order to consider it a negligible difference, or “consistent” as defined in this paper?’**: To quantitatively study convergence with width, in Figure 3 of the rebuttal pdf we fit $C, D, \beta$ in the equation $L = C (width)^{-\beta} + D$ via the scipy curve_fit function. We find an excellent fit to the observed data for both language and vision tasks. This is consistent with diminishing returns of increasing width. We plotted this for a given training step near the end of training, but consistently observe the same power-law trends in $N$ throughout training.
> 5. **‘Ensembling in logit plots’**: Indeed we are averaging those plots over initialization. We will make sure to add this to the discussion along with error bars showing the standard deviation.
> 6. **'Benign overfitting on test'**: Both bias ($(\left< f_N \right> - f_\infty)^2$, as discussed above) and variance tending to 0 for test predictions implies that test predictions are converging with the widths used in the experiment. On the other hand this does not happen for train where larger widths overfit more i.e. more overfitting on train at larger widths does not lead to worse test performance. This is why this is an instance of benign overfitting. We will expand the discussion to make this clear.
>
>
> We thank the reviewer for pointing out that our language was confusing in Section 4. We will make our explanations more concise and clear. Below we answer some of the questions
>     1. We will fix this sentence, writing instead that  "... although the eigenvalue spectrum of the ensembled eNTK is not substantially affected by finite width, the correlation of the top eigenfunctions with the target function decreases with width."
>     2. The "task" is simply fitting the target polynomial $y(x)$ in the online setting. We will remove the use of "task" for $y(x)$ in the text.
>     3. We wanted to identify a setting where a simple explanation could account for the bias gaps across widths. For gradient flow, the bias correction has to do with changes in the statistics of the NTK $K$ across widths. In principle, the eigenvalues and eigenvectors of $K$ could change, but we found that the difference in the dynamics was largely due to changes in the eigenvectors.
>     4. We will use a simpler vernacular. Instead of referring to decompositions and "spread" of power, we will switch to using "correlation" language. For instance, the correlation of eigenfunction $k$ with the target is $\left< y(x) \psi_k(x) \right>$. The test MSE depends on these correlations and on the eigenvalues $L = \sum_{k} \left< y(x) \psi_k(x) \right>^2 e^{-2 \lambda_{k} t}$. We will then refer to $C(k)$ as the correlation of $y$ with the top $k$ subspace of eigenfunctions $C(k) = \sum_{\ell=1}^k \left< \psi_k(x) y(x)\right>^2_x$. This metric is merely meant to capture the alignment of the top eigenvectors with the target function.
>
>
> **'figures/fonts in figures are far too small'**:  We will make sure to correct this by increasing the font and figure size (we will have one extra page for the final version).
>
> **'missing intuitions, explanations, contextualization'**:  We will make sure to do this in the final version (we will have one extra page for the final version). If there is a specific explanation that the reviewer finds missing we would be happy to provide it now.

---

### Official Review · Reviewer_g9eH · 2023-07-09

**Soundness:** 3 good
**Presentation:** 3 good
**Contribution:** 3 good
**Rating:** 5
**Confidence:** 3

**Summary:**

The paper investigates the impact of width on the dynamics of feature-learning neural networks with both vision and language tasks. The empirical results demonstrate that during the early stage of training or during the entire training process for easy tasks, wide neural networks trained on online data not only exhibit identical loss curves but also display consistent point-wise test predictions. Nevertheless, for more challenging tasks, in the later stage of training, deviations across finite widths become more pronounced. The authors then decompose the deviation to bias and variance and show that the variance scales inversely with width. Furthermore, the authors demonstrate that ensembles of narrow networks can have inferior performances compared with a single wide network.

**Strengths:**

1. The paper is well-organized and clearly presented.
2. All the experimental results are clearly motivated and effectively support the claims made in the paper.
3. The supplementary material is comprehensive, providing extensive details and codes for numerous experiments conducted in the study, enhancing reproducibility and transparency.

**Weaknesses:**

he reviewer's major concern about the paper is the potential implications of the paper. While the empirical results presented in the study are well-documented, it remains challenging to discern how this work motivates or contributes to the broader research field. In the conclusion section, the authors assert that the findings "motivate the applicability of infinite-width feature-learning models in reasoning about large-scale models trained on real-world data." However, the empirical results within the paper indicate that deviations caused by varying widths amplify for more complex tasks or longer training time. Moreover, there is no apparent evidence suggesting that increasing width follows the law of diminishing marginal utility. Consequently, the reviewer struggles to grasp how the results presented in the paper effectively support the applicability of infinite-width models.

**Questions:**

In section 3, the authors decompose the deviation into bias and variance, how did the authors quantify the two terms? It seems like bias is represented by the training loss, and variance is quantified by the improvement brought by ensemble?


**Limitations:**

Yes.

---

> ### Author Rebuttal · Authors · 2023-08-09
>
>
> We thank the reviewer for their comments. We are glad that the paper was received as well-organized with well-motivated results.
> We would like to address the main concern: the broader applicability of our findings.
>
> * *"In the conclusion section, the authors assert that the findings 'motivate the applicability of infinite-width feature-learning models in reasoning about large-scale models trained on real-world data.' However, the empirical results within the paper indicate that deviations caused by varying widths amplify for more complex tasks or longer training time."*
>
> We agree that a large part of our results analyze the leading deviations from infinite width behavior rather than showing it holds ubiquitously. We are thus happy to change the final sentence to something that more accurately represents the conclusions of the paper. We have modified the sentence to
>
> "These studies motivate the study of infinite-width feature-learning models, together with their leading deviations, as a promising avenue for reasoning about the early training periods of large-scale models trained on real-world data."
>
> Despite the amplifications at later training times, we have consistently found that at any given fixed training time, wider networks converge to a limit, and sufficiently large networks are consequently quite close to the infinite width limit. The widths required were computationally accessible at late training times even in the realistic settings of ImageNet and Wikitext. Even when the network deviates from the infinite width limit at later times, the leading order deviations can still be systematically studied. Since the writing of this paper, infinite width feature-learning theory was extended to handle finite width deviations [1].
>
> Further, for simpler tasks commonly used as benchmarks for deep learning theory, the infinite width limit is quite easily reached for moderate widths and persists as a valid approximation throughout training.
>
>
>
> * *Moreover, there is no apparent evidence suggesting that increasing width follows the law of diminishing marginal utility*
>
> To address this comment, we have included a set of new plots. Figure 5 in the rebuttal PDF consider the loss of a muP-scaled ResNet-18 model on ImageNet after 2 epochs as a function of $N$. Across all training times we find diminishing returns to increasing the width. We also show diminishing returns as a function of the size of the ensemble in the same panel of figures.
>
> Going beyond this, we have also added Fig 3 which plots an explicit fit to the loss as a function of width. We fit $C, D, \beta$ in the equation $L = C N^{-\beta} + D$ via the scipy curve_fit function. We find an excellent fit to the observed data for both language and vision tasks. This is consistent with diminishing returns of increasing width. We plotted this for a given training step near the end of training, but consistently observe the same power-law trends in $N$ throughout training.
>
> * *the reviewer struggles to grasp how the results presented in the paper effectively support the applicability of infinite-width models.*
>
> In Fig 1 of the supplementary material, we plot a network trained on a vision task in both the $\mu$P and the NTK parameterizations. The NTK trained network does not converge to the infinite width NTK behavior over the given set of widths, while the $\mu$P networks' loss curves become consistent and closely match. Although infinite-width feature-learning networks are challenging to simulate, it does appear that realistic networks can approach their behavior far more easily than NTK-parameterized networks can reach the NTK.
>
> We hope that these new studies, together with the refinement of the conclusion, prove to be more convincing.
>
> **Response to Question:**
>
> We agree that it would be better to be far more explicit in the text about the meaning of the terms "bias" and "variance". In the final draft, we will always preface them with "initialization bias" and "initialization variance".
>
> We will clarify what we mean by *bias* and *variance*. By bias, we merely refer to the deviation between the ensembled predictor and infinite width predictor $(\left< f_N \right> - f_\infty)^2$. For the MSE between $f_N$ and $f_\infty$, we have the typical bias variance decomposition $\left< (f_N-f_\infty)^2 \right> =  (\left< f_N \right> - f_\infty)^2   + \left<(f_N - \left< f_N \right>)^2 \right>$.
>
> We have added a section in the appendix that now includes the discussion in the above paragraph and adds more detail. We hope this allows for readers to understand how our notions of bias and variance are quantitatively analogous with the standard ones in statistical learning. We also have added a reference to a paper that we believe most clearly elaborates on such a decomposition [4].
>
> If there are further concrete changes that you would like us to make, or additional experiments to run, we would be happy to include them in the final draft.
>
> [1] Blake Bordelon, Cengiz Pehlevan. Dynamics of Finite Width Kernel and Prediction Fluctuations in Mean Field Neural Networks. 2023
>
> [2] Mario Geiger, Arthur Jacot, Stefano Spigler, Franck Gabriel, Levent Sagun, Stéphane d’Ascoli,  Giulio Biroli, Clément Hongler, and Matthieu Wyart. Scaling description of generalization  with number of parameters in deep learning. 2020
>
> [3] Alexander Atanasov, Blake Bordelon, Sabarish Sainathan, and Cengiz Pehlevan. The onset of variance-limited behavior for networks in the lazy and rich regimes. 2022
>
> [4] Ben Adlam and Jeffrey Pennington. Understanding double descent requires a fine-grained bias-variance decomposition. 2020

---

> > ### Comment · Reviewer_g9eH · 2023-08-17
> >
> > Thanks to the authors for the detailed response and for clarifying my concerns, I will thus increase my score.

---

### Official Review · Reviewer_gHF9 · 2023-07-24

**Soundness:** 3 good
**Presentation:** 3 good
**Contribution:** 3 good
**Rating:** 6
**Confidence:** 4

**Summary:**

This is a primarily empirical paper studying the agreement between real, finite-width neural networks and their limiting behavior under infinite width. Rather than using the neural tangent kernel as the descriptor of infinite-width behavior, the paper focuses on a different parameterization called feature-learning networks, which vary from standard-parameterization networks in that intermediate layer weights do not remain fixed even in the infinite-width limit. For these feature-learning networks trained in the online setting (without repeating training data) across a mixture of architectures and vision and language tasks, the paper demonstrates that large but finite-width networks tend to agree across a wide variety of metrics including overall loss, per-sample predictions, and the distribution of learned weights. However, if networks are too narrow, or are trained in the offline setting (when data is repeated during training) this agreement can break down. This gap between finite and infinite-width performance is then studied using the empirical NTK (at initialization) as well as the empirical after-kernel, and the results suggest that finite width induces deviation in the kernel eigenvectors (but not its eigenvalues) compared to the infinite-width limit.

**Strengths:**

I really appreciate the thorough and systematic experimentation. Deep learning theory benefits from the ability to model infinite-width networks, but it’s important to know how well and under what circumstances such infinite-width limits translate into finite-width practical behaviors. In particular, I appreciate that this paper includes a mixture of convolutional and attention-based neural networks on a mixture of vision and language tasks, and studies finite and infinite-width agreement across loss, per-sample predictions, and network weights. It also finds some interesting and useful boundaries for when we can expect agreement to hold versus when finite-width networks behave differently, and takes steps to explain why.

**Weaknesses:**

Most of the weaknesses I’ll describe should be viewed as suggestions to improve the exposition of the paper, and in some cases suggestions to make the experiments clearer or more compelling. I tried to separate suggestions/complaints into this section and clarification questions into the “questions” section, but there isn’t always a perfect distinction.
- The abstract uses the term “feature-learning neural networks” without definition. Thought it’s defined early on in the introduction, many potential readers may be confused by the abstract before they see the definition in the introduction.
- There are a decent number of typos and small grammatical mistakes, though they don’t impede comprehehension. For example, Figure 1 caption says “Loss curves…are nearly to identical at large widths.” Line 37 says “We put all of our in mu-parameterization”—similar missing-word type typos appear throughout the text.
- Essentially all of the figures require the reader to zoom in to understand what is being shown. Figures should be fully interpretable without zooming in, e.g. for readers who use a printed copy. Axis labels, legends, tick labels, and line widths all need to be substantially larger.
- Lines 122-125 define a bunch of notation that is (1) confusing, especially for readers unfamiliar with Transformers, and (2) is never referenced again. Instead, figure 2 uses “d” without any subscript, which is not defined. It would be preferable to define (only) the necessary notation, and push other details to the supplement.
- Figure 3 is a bit confusing. It’s not clear if the first row is literally showing results from a single test point, or if it’s an average over test points. I’m also not sure why these figures only consider the correct logit value as a metric of agreement; I would have thought a more natural comparison to be some metric of model agreement across predictions, such as overlap in where mistakes are made (used in https://arxiv.org/pdf/1905.12580.pdf).
- Figures 4c and 4f and 8c and 8f all plot 1-{some quantity} on the y axis, whereas in the textual description trends are described with respect to {some quantity}. This is needlessly confusing.
- Line 157 describes figure 5a as growing “steadily to a final value that it then fluctuates around” but in the figure I don’t really see a stable final value with fluctuations; rather it seems the trend continues to increase even after fluctuations arise. This is a minor point, but was a bit confusing since it otherwise seems the main purpose of the plot is to visualize when the fluctuations begin.
- Figure 7: Please label the x axis with epochs either instead of or ideally in addition to training steps. The purpose of the figure is largely to investigate the effects of reusing training data from one epoch to the next, so it’s important to see when data is first reused. For example, in subfigures (a) and (b) I suspect this happens around step 10000 when the curves diverge, but if this is the case I wonder why not include more training steps so that we can see the deviations more clearly?
- Equation 1 seems to be describing the NTK. I wonder why the preceding paragraph that describes its derivation doesn’t mention NTK? As I read that paragraph I kept thinking “this sounds just like NTK” but I wasn’t sure why the definitions were present unless either it’s different in some way from NTK (in which case, please specify the difference directly) or it’s a refresher for readers who aren’t familiar with NTK (in which case, please say so to avoid ambiguity).


**Questions:**

- Section 3 describes ensembling of finite-width models as a way to reduce variance, but shows that ensembled smaller models are still worse than a single larger model. How did you decide how many smaller models to ensemble? In particular, I wonder if there is some combinatorial flavor here (inspired by https://arxiv.org/abs/1803.03635) where an exponential number of smaller models is needed.
- In Section 3.2, what motivates the use of these two datasets? In particular, I would think comparisons of the effect of label noise would be easier to see if using the same dataset with vs without label noise, rather than two different datasets where one of them happens to be noisy.
- In Section 4, why use NTK and after-kernel? I presume this is because there are not similarly computable limits for feature-learning networks? In general I wonder why the paper uses large but finite models as a proxy for infinite-width networks, if feature-learning limits are well understood. An explanation of this reasoning and more detail on related work for what is known about the infinite-width limits of feature learning networks would be valuable.

**Limitations:**

Limitations of the experiments are not directly discussed, and the paper would benefit from such a discussion, perhaps in the conclusion. One limitation would be the lack of theoretical or direct modeling of the infinite-width limit, and the use of large-width networks as a proxy.

---

> ### Author Rebuttal · Authors · 2023-08-09
>
> We thank the reviewer for their useful comments and suggestions. Below we address the various weaknesses and questions which arose in the review.
>
> ### Weaknesses
>
> 1. We will include a sentence in the abstract specifying that we mean networks in $\mu$P. We agree that we should define what we mean by "consistency of feature learning networks" earlier on.
> 2. Thank you for finding these incomplete sentences and grammatical errors. We will comb through the text and check for these errors.
> 3. We will increase fontsize and figure size using the additional space, provided that our paper is accepted.
> 4. We will define $d$ as the embedding dimension used for the queries and key vectors. We wanted to be explicit what our transformer sizes were for readers whose primary interest was language models. We can move this discussion to the appendix if it confuses or detracts. The $d$ in Fig 2c is indeed confusing and we will replace it with $N$.
> 5. Figure 3 is a *single test example* from the test set but the curve is averaged over the random initial networks. We will clarify this in the new version.
> 6. We will try avoiding plotting (1 - {quantity}) in our plots. Either by redefining the metric so it is just {quantity} on the y axis or just plotting the current {quantity}. We preferred 1-CKA to CKA since it lets you see the scaling law of convergence of the kernels on a log-log plot.
> 7. In plots involving multiple passes over the data, we will make the $x$ axis epochs. We agree this will allow us to see the effect of multiple passes over the same data. In the plots in the paper and the attached pdf data actually starts to repeat much earlier, at 400 steps! (though there is data augmentation) So 10s of epochs are needed for overfitting to emerge.
> 8. Equation 1 is definitely the NTK! We are sorry this was not stated clearly, we only say "the kernel" but what we really mean is the "neural tangent kernel" for the NN.
>
> ### Questions
> 1. We experimented with different ensemble numbers and verified that the results were stable under addition of new ensembles. See Figure 5 left panel in attached rebuttal PDF for Imagenet experiment with varying number of ensembles. The curves quickly converge in $E$. For MSE loss it is easy to theoretically show that the loss decays as $B + S/E$ where $B$ and $S$ are fixed constants.
> 2. Both of the experiments in Section 3.2 are just subsamples of CIFAR-5M. They were both chosen so that the data could be passed over multiple times. For the noisy data, we saw a more dramatic overfitting effect if the dataset was smaller. We can provide plots for other subsamples if the reviewer is interested.
> 3. In section 4, we look at the final NTK measured from data since we do not have a computationally tractable way to compute the infinite width limit or its finite size deviations. Empirically other works have shown that the after kernel does seem to capture important properties of the final model (Long 2021 arXiv:2105.10585). We will add additional information about what is known of the infinite width feature learning limit and why it is challenging to compute.

---

> > ### Comment · Reviewer_gHF9 · 2023-08-13
> >
> > Thanks to the authors for the thorough response. I continue to lean in favor of acceptance.

---

> > > ### Author Response · Authors · 2023-08-15
> > >
> > > We thank the reviewer for their continued support and for appreciating our responses to their questions. We are interested to hear from the other reviewers to see if their questions have also been addressed

---

### Official Review · Reviewer_ddkm · 2023-07-26

**Soundness:** 2 fair
**Presentation:** 2 fair
**Contribution:** 3 good
**Rating:** 6
**Confidence:** 3

**Summary:**

In this work, the authors conducted an empirical investigation into the consistency across widths of the training dynamics of neural network models in the maximum-update scaling. Specifically, they demonstrated through experiments on both vision and language tasks that many behaviors in the online learning dynamics are consistent at large widths, including the loss values, single predictions, feature kernels and large-learning-rate phenomena, and that ensembling improves the performance. Deviations from the large-width behavior are reported in the offline learning setting. Finally, the authors argued through heuristics and experiments that the finite-width bias is largely due to the deviation in the tangent kernel's eigenvectors.

**Strengths:**

Given the number of theoretical results concerning over-parameterized neural networks and their infinite-width limits, the question of the finite-width deviation from the infinite-width limit is an important one, especially under the feature-learning regimes. To my knowledge, this is the first work to put forth an empirical investigation of the large-width consistency of neural networks in the maximum-update parameterization. The authors of the paper performed a wide range of numerical experiments on various tasks with different neural network models that cover several relevant aspects of the model training behavior.

**Weaknesses:**

Note that for feature-learning neural networks in an alternative (mean-field) parameterization, both empirical and theoretical studies of similar purposes already exist, with examples including Nguyen (2019), Chen et al. (2020) and Geiger et al. (2020). More comments below regarding parameterizations.

The authors argued based on Figures 2 and 7 that, as the widths of the model increase, the training performance of the single network converges in the online learning setting but not in the offline setting, which I don't find fully convincing. The y-axis uses a linear scale in Figure 2 and logarithmic scale in Figure 7, and hence the discrepancy of the losses across different widths in Figure 7 (which appears to be quite large late in training under the logarithmic scale) may not actually be much more significant than in Figure 2. In addition, it is also hard to rule out the possibility that the width being used is not yet large enough to approximate the large-width limit (after all, 512 is still small compared to the size of the datasets).

The original proposal of the maximum-update parameterization does not include normalization layers, and given their role in preserving the scale of the hidden-layer features, one would expect their effect to interact with the initialization scale of the parameters. The authors only mentioned in Appendix A.2.1 that LayerNorm was used in the experiments on CIFAR-5m and did not comment on whether normalization layers are used in the other experiments. I think the authors should discuss in detail whether normalization layers were included in the other experiments and, if so, why the phenomena observed in this work can be truly associated with the maximum-update parameterization.

References:

1. Nguyen, "Mean Field Limit of the Learning Dynamics of Multilayer Neural Networks", arXiv 2019

2. Chen et al., "A Dynamical Central Limit Theorem for Shallow Neural Networks", NeurIPS 2020

3. Geiger et al., "Scaling description of generalization with number of parameters in deep learning." Journal of Statistical Mechanics: Theory and Experiment, 2020

**Questions:**

(Please see the comments above.)

---

> ### Author Rebuttal · Authors · 2023-08-09
>
> We thank the reviewer for suggesting these important related works. We will be sure to include a proper comparison of our study to these previous papers. Below, we provide a brief comparison of our results to each prior work.
>
> #### Comparison to prior works
> 1. First [1] Chen et al study two layer neural networks in the mean field parameterization and establish a central limit theorem where the mean of the neural network predictor dynamics is the infinite width limit and the fluctuations scale inversely with width. However, this finding does not carry over to deeper networks, where the average of finite width networks deviates from the infinite width limit (what we call the bias correction of finite width). This gap between ensembled narrow networks and wider networks observed in our work demonstrates that ensembling many finite width networks is not equivalent to the performance of an infinite-width deep network, even with infinite ensembles. Chen et al also provide experiments showing lawful fluctuations that scale as $1/N$ but have similar dynamics in student-teacher settings and a simple non-planted two dimensional target function. We will be sure to cite this paper as evidence that fluctuations and average dynamics can be consistent for large but finite $N$.
> 2. Second [2] Nguyen et al study a slightly different parameterization which also gives an infinite width feature learning limit. To see the difference between our parameterization and theirs, note the factor of $1/N$ in equation 7 of [2] for the intermediate layer, which would be $1/\sqrt{N}$ in our paper. Nguyen et al do provide experiments which show width consistent dynamics in their parameterization for MNIST and CIFAR-10 in fully connected networks with a handful of layers $\sim 5$. This parameterization behaves differently at infinite width compared to the parameterization we study. Concretely, in another work (Nguyen & Pham 2020 arXiv:2001.11443), Nguyen and Pham show that iid random initialization leads to a degenerate limit for any depth $L \geq 4$, which makes it impossible for successful finite width training from iid init to be consistent with infinite width networks. On the other hand, the $\mu$P infinite width limit is nondegenerate for any finite depth $L$ with iid initialization. This makes it *possible* that finite width networks could be consistent with their infinite width dynamics and motivates an empirical study of finite width consistency. Further, our work provides experiments for deeper networks with a variety of architectures and datasets including Residual CNNs (depth $\sim 20$) and transformers ($\sim 4$ encoder blocks which is like $\sim 12$ total layers) and we document consistency for other relevant deep learning phenomena (feature kernels, preactivation distributions, large learning rate effects etc).
> 3. Geiger et al [3] provide a detailed study of NNs trained on NTK parameterization, where the large width limit is given by a kernel method. Their study also documents interesting fluctuation effects from initialization variance and have experiments on MNIST and CIFAR-10, convincingly arguing that corrections to infinite width scale as $1/N$ due to finite size fluctuations. Our study is different from this study in that we provide $\mu$P networks which have a feature learning infinite width limit and we focus on dynamics of many observables during learning, rather than the error curve for fully trained networks. A different work from Geiger and colleagues (which we currently cite) does study the dynamics as a function of feature learning strength and characterize a parameterization that is equivalent to $\mu$P as a limit where width consistent feature learning is obtained (Geiger et al 2019 arXiv:1906.08034).
>
> #### Layernorm
> We thank the reviewer for this question and allowing us a chance to clarify our use of LayerNorm. In the original submission, the ResNets on CIFAR-5m and ImageNet and the transformers trained on Wiki-text all utilized LayerNorm. The experiments on MLPs in Section 4 do not have LayerNorm.
>
> Layernorm does not prevent a well defined infinite width limit in $\mu$P. Several works have established the infinite width signal propagation of networks with Layernorm (Doshi et al 2021 arXiv:2111.12143, Yang 2020 arXiv:2006.14548, Yang & Hu 2021 arXiv:2011.14522). The key idea is that preactivation vectors (before layer norm) $h^\ell \in \mathbb{R}^N$ will have means $u^\ell = \frac{1}{N} \sum_{i=1}^N h^\ell_{i}$ and variances $V^{\ell} = \frac{1}{N} \sum_{i=1}^N ( h^\ell_{i} )^2 - ( u^\ell )^2$ that concentrate in the $N \to\infty$ limit. These objects $\bar{h}, v$ are covered by Yang's Master Theorem (Theorem 7.4 here arXiv:2011.14522).  Thus at infinite width, LayerNorm will merely shift and divide each preactivation vector by a deterministic value $h^\ell_{\mu i}  \to \left(h^\ell_{\mu i} - \bar{h}^\ell_{\mu} \right) / \sqrt{v^\ell_\mu}$. Backpropagation signals can similarly be reasoned about. Frameworks like Tensor Programs (Yang & Hu 2021 arXiv:2011.14522) or Dynamical Mean Field Theory (Bordelon & Pehlevan 2022 arXiv:2205.09653) could be used to more carefully demonstrate how $u^\ell_\mu(t), v^\ell_\mu(t)$ will concentrate and evolve over training time $t$. We will comment on this in the Appendix.
>
> In addition, to further test that the width consistency we find is not an artifact of LayerNorm, but is also visible in $\mu$P networks without layernorm, we provide an example experiment on CIFAR-5m. In the rebuttal PDF we show an experiment of many networks trained without LayerNorm showing width consistency (Figure 4).

---

> ### Author Response · Authors · 2023-08-18
> **Discussion**
>
> We would also like to take this discussion period as an opportunity to highlight our main rebuttal post as it relates to this review. We have added Figures 2,3 in the supplemental material to address the reviewer’s comments on
> * The choice of log scale for the offline setting
> * The question of whether the networks are indeed converging to their infinite width analogues.
> In Figure 2, we have re-plotted the data using a linear y-axis scale. The deviation between the offline and the online learning curves is still quite noticeable beyond $2^{13}$ steps. If both axes were made linear, this deviation would be accentuated further still. In Figure 3 we show that a power law of the form $D + C N^{-\beta}$ provides an excellent fit to the losses at a given step near the end of training as a function of width. This regularity provides evidence that we may expect to reliably extrapolate the trained networks’ learning curves to infinite width behavior (as given by $D$). We will be sure to update the scale in Figure 7 of the main paper and add in a plot of the power law curves to our final draft.
>
> In addition, we have Figures 1 and 4 in the rebuttal pdf which show training dynamics without layernorm, showing that the results are not sensitive to layernorm.
>
> We are happy to clarify any of these claims and are eager to hear any further comments that the reviewer has.

---

> > ### Comment · Reviewer_ddkm · 2023-08-21
> > **Response to the authors' rebuttal**
> >
> > I thank the authors for the detailed response and additional results! I agree that the subject treated in this paper has clear novelty compared to the three earlier pieces of work mentioned in my original review. It is also helpful that the authors clarified the effects of LayerNorm in the experiments, and the authors' argument that an infinite-width limit can still be defined with LayerNorm is quite reasonable.
> >
> > Regarding the doubt I expressed earlier regarding whether there truly is a qualitative difference between the offline and the online settings in terms of the large-width behaviors: I like the new plot that compares the two settings under a single plot with a linear scale for the y-axis. However, while the difference in loss values across widths seems indeed larger in the offline setting late in training, I am not certain if any qualitative conclusions can be directly drawn from it. After all, as mentioned in my original review, the widths are still limited compared to the size of the data set.
> >
> > Overall, considering the clarifications and improvements that we see from the authors' rebuttals, I would like to raise my score.

---

### Author Rebuttal · Authors · 2023-08-09

We thank the reviewers for their detailed reading of our paper and useful suggestions and comments. In response to many repeated questions and concerns, we have performed the following new experiments (see attached PDF)

#### New Figures and Experiments
1. We have added a comparison of network dynamics for $\mu$P and NTK networks (without layernorm). While the $\mu$P networks quickly approach a consistent limit, the NTK networks exhibit different dynamics and train more slowly as width increases. Width consistency is harder to achieve in NTK parameterization (or standard param with learning rate $1/N$) since (a) kernels must concentrate and (b) feature learning effects must go to zero. Consistency in $\mu$P is a strictly weaker condition as it only requires (a) concentration. We will stress this point more in our our paper.
2.  We verify that layernorm does not significantly change our results, showing strong width consistency on CIFAR-5M. We also provide a theoretical argument that layernorm should not prevent a well defined infinite width limit, and argue that this falls under the Master Theorem of Yang & Hu 2021 (see response to reviewer ddkm).
3. We plot the explicit scaling relationship for both Imagenet models and Transformer models at a fixed time, showing that a power law fits the data very well.
4. We train a myrtle network without layernorm on CIFAR-5m and observe similar consistency in that setting as well.
5. (5a) We provide plots that vary the number of ensembles to verify that we have used a large enough ensemble count (20) to estimate average loss and bias.
5.  (5b) Similarly, we show that returns to making network wider exhibit diminishing returns by plotting performance at a fixed time as a function of $N$.
5. (5c) We check whether zeroing out the last layer at initialization has a significant effect on the dynamics/consistency, and find little change.

#### Writing updates
In addition to the new experiments we also aim to make some parts of our paper more clear. Specifically, we hope to clarify the following points
 1. We will provide more citations and connections between our work and existing empirical papers which study width consistent dynamics (see Reviewer ddkm). Some prior works are concerned with similar effects (eg fluctuations due to finite size). We think our paper is novel in that we have stress tested the infinite feature learning limit in experiments on larger scale networks and datasets and in the $\mu$ parameterization and examined many metrics including internal losses, representations, large step size effects, etc (see response to ddkm).
2. We will clarify what we mean by *bias* and *variance*. By bias, we merely refer to the deviation between the ensembled predictor and infinite width predictor $(\left< f_N \right> - f_\infty)^2$. For the MSE between $f_N$ and $f_\infty$, we have the typical bias variance decomposition $\left< (f_N-f_\infty)^2 \right> =  (\left< f_N \right> - f_\infty)^2   + \left<(f_N - \left< f_N \right>)^2 \right>$. We note that this is not the only type of bias/variance decomposition used in Machine Learning theory (Adlam & Pennington arXiv:2011.03321).
3. We will explain why we resort to experiments to try establishing convergence to infinite width rather than computing the infinite width theory. Computing the feature learning infinite width limit is challenging (super exponential time for exact numerical results, or $\sim T^3$ time for $T$ steps of SGD with Monte-Carlo methods).
4. Since finite size effects do tend to accumulate in large scale settings and this is an important aspect of our empirical results, we change our sentence in the discussion from "our results motivate studying infinite width feature learning limits" to "our results motivate studying infinite width limits and the accumulating finite width deviations from this limit."
5. We will acknowledge our use of our widest networks as a proxy for the infinite width limit as a limitation of our results in the discussion.
6. We will make the theoretical section of our paper more clear and readable and invoke minimal jargon. With the additional page of space, we can provide longer explanations of what we aim to establish there (explaining the source of bias gaps in a simple setting). We will use a more standard vernacular and avoid convoluted phrases about "spread of task power" etc.
7. We will be clear about how we are averaging our plots over random seed and will add error bars for standard deviation over inits where applicable. We will also make the fonts and figures bigger so that they are more readable.
8. We will clarify that $\mu$P should not be thought of as a different type of network than finite width standard/NTK parameterization networks, but rather as a rule for how to make a finite width SP network wider without sacrificing feature learning (preventing a limiting kernel behavior or unstable dynamics). See figure 1 in attached PDF for a clarification.

---

### Decision · Program_Chairs · 2023-09-21

**Decision:**

Accept (poster)

**Comment:**

The paper shows realistic networks achieve infinite-width feature-learning behavior throughout training for easy tasks and for early training time in hard tasks. The authors conducted an empirical investigation into the consistency across widths of the training dynamics of neural network models in the maximum-update scaling. Rather than using the neural tangent kernel as the descriptor of infinite-width behavior, the paper focuses on a different parameterization called feature-learning networks, which vary from standard-parameterization networks in that intermediate layer weights do not remain fixed even in the infinite-width limit.

The reviewers pointed out this work provides strong, novel empirical study. The experimental  coverage is wide enough to extract practical applicable lessons. Some notable quotes from the reviewers:

- "the first to consider the empirical investigation of the large-width consistency of neural networks in the maximum-update parameterization" (`ddkm`)

- "a wide range of numerical experiments on various tasks with different neural network models that cover several relevant aspects of the model training behavior." (`gHF9`)

- "appreciate that this paper includes a mixture of convolutional and attention-based neural networks on a mixture of vision and language tasks, and studies finite and infinite-width agreement across loss, per-sample predictions, and network weight" (`gHF9`)

Also during rebuttal authors added empirical comparison of network dynamics for muP and NTK parameterized networks with regards to width consistency and further demonstrated weak dependence on inclusion of LN.

With growing interest in the feature learning limit of deep learning, broad empirical study on consistency of width in these limits is valuable for both practical applicability and further theoretical study in the future. As reviewers unanimously recommended accepting the paper (6, 6, 6, 6, 5), AC recommends acceptance.